# `DOTA`: DistributiOnal Test-time Adaptation of Vision-Language Models

**Zongbo Han**[1,2*], **Jialong Yang**[3], **Guangyu Wang**[1,2], **Junfan Li**[4], **Qianli Xu**[5],
**Mike Zheng Shou**[6*], **Changqing Zhang**[3*]
School of Information and Communication Engineering, Beijing University of Posts
and Telecommunications[1]
State Key Laboratory of Networking and Switching Technology, Beijing University of
Posts and Telecommunications[2]
College of Intelligence and Computing, Tianjin University[3]
School of Computer Science and Technology, Harbin Institute of Technology Shenzhen[4]
Institute for Infocomm Research (I[2]R), A*STAR Research Entities (ARES)[5]
Show Lab, National University of Singapore[6]

## Abstract

Vision-language foundation models (VLMs), such as CLIP, exhibit remarkable performance across a wide range of tasks. However, deploying these models can be unreliable when significant distribution gaps exist between training and test data, while fine-tuning for diverse scenarios is often costly. This creates a need for methods that can efficiently adapt to new data at test time without expensive retraining. Cache-based test-time adapters serve this purpose by storing representative test samples to guide subsequent classifications. Yet, these methods typically employ naive cache management with limited capacity, leading to severe catastrophic forgetting when samples are inevitably dropped during updates. In this paper, we propose `Dota` (DistributiOnal Test-time Adaptation), a simple yet effective method addressing this limitation. Crucially, instead of merely memorizing individual test samples, `Dota` continuously estimates the underlying distribution of the test data stream. Test-time posterior probabilities are then computed using these dynamically estimated distributions via Bayes' theorem for adaptation. This distribution-centric approach enables the model to continually learn and adapt to the deployment environment. Extensive experiments validate that `Dota` significantly mitigates forgetting and achieves state-of-the-art performance compared to existing methods. Code is available at https://github.com/skylineeeeen/DOTA.

## 1 Introduction

Recent advances in vision-language foundation models have shown remarkable vision understanding capabilities across a broad range of tasks by training on web-scale image-text pairs [39, 30, 51]. Taking CLIP as an example, it can conduct zero-shot classification without the need for additional training data using predefined prompts [39]. However, CLIP may still face challenges when handling various specific applications during test time, especially when there is a significant distribution gap between the training and test data [41, 25, 13]. To adapt a foundational model to diverse deployment environments and personalized application requirements, fine-tuning is often necessary, which can be resource-intensive in terms of time, computational effort, and training data requirements.

Test-time adaptation (TTA) methods provide an efficient solution for addressing distributional shifts between training and testing domains [5, 6, 45]. TTA allows for the dynamic adjustment of a pre-

---

*Corresponding author: zongbo@bupt.edu.cn, mike.zheng.shou@gmail.com, zhangchangqing@tju.edu.cn

39th Conference on Neural Information Processing Systems (NeurIPS 2025).

trained model during the inference phase by leveraging incoming test data to refine the model's parameters. This adaptation process optimizes the model's performance for the specific test distribution, eliminating the need for resource-intensive retraining. As such, TTA aligns seamlessly with real-world applications, where models must rapidly adapt to diverse and changing environments. There are two primary lines to achieve TTA on VLMs. Early works advocate learning prompts during test time with the test data [41, 13]. However, these methods require significant computational resources to optimize the learnable prompts via backpropagation and gradient descent, rendering them unsuitable for applications demanding fast test-time inference. Therefore, more efficient methods, cache-based methods, have been proposed recently [25, 55]. Typically, to avoid the need for training with backpropagation, Training-free Dynamic Adapter (TDA) maintains a lightweight cache during testing to store representative test samples, which helps guide the classification of subsequent test samples.

While cache-based classifiers like TDA offer significant efficiency, they are fundamentally constrained by their finite cache capacity. These methods typically store a limited set of 'typical' test samples, updating the cache by replacing older entries with newer, high-confidence ones. However, this reliance on instance-level memorization and forced sample discarding inevitably leads to catastrophic forgetting. As the model adapts to new data patterns, it loses information about previously seen variations, hindering the formation of a stable and comprehensive understanding of the evolving test distribution. Consequently, classifiers depending solely on cached samples can be suboptimal, and ultimately performance-limited by the cache size [55]. As illustrated in Fig. 1, to overcome these limitations of instance-level caching, we introduce DistributiOnal Test-time Adaptation (Dota). This represents a paradigm shift in test-time adaptation. Instead of passively memorizing discrete samples in the cache with limited size, the core motivation of Dota lies in actively and continuously estimating the underlying statistical distribution of the incoming test data, thereby enabling the full utilization of all available test samples.

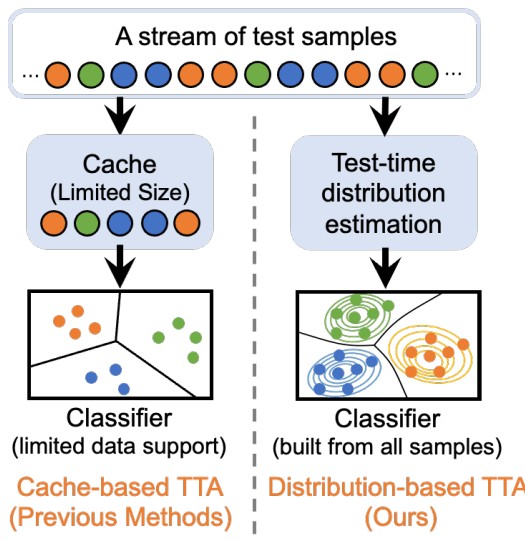

Figure 1: Cache-based TTA methods store individual test samples within a limited cache, which often leads to underutilization of the available test data. In contrast, the proposed method continuously estimates the underlying distribution of the test data, enabling the full exploitation of all available test samples.

Specifically, Dota continually estimates the distribution of test samples to adapt the test environment. Under the mild assumption that the embedding distribution of each class follows a Gaussian distribution [19], we propose an efficient method to continually estimate the distribution of different classes. Once the distributions of different classes are estimated, we can easily calculate the posterior probabilities of subsequent test samples based on Bayes' theorem and obtain a test-time classifier for test-time adaptation. Similar to cache-based methods, this process does not require gradient backpropagation, avoiding the complex computational overhead during testing, leading to more than 20 times faster inference speed. Moreover, unlike cache-based methods memorizing only representative test samples, Dota can continually adapt to the test environment by estimating the distribution of test samples. The contributions of this paper are:

- We propose a novel continual test-time learning framework which improves the performance of pretrained foundation models in downstream tasks by learning the statistical distribution of test data, rather than caching a limited set of individual samples.

- Within this framework, we propose a simple and effective method to enhance foundation models by estimating distribution of different categories at test time and using Bayes' theorem to create an adaptive classifier.

- Extensive experiments on diverse datasets validate the effectiveness of the proposed method, demonstrating a significant improvement. The code has been released at here.

## 2 Related work

**Test-time adaptation (TTA)** for classical classification neural networks focuses on addressing the distribution shift between training and test data by learning from the test data. Early efforts to improve TTA performance primarily involve adjusting batch normalization layers and designing unsupervised objective functions [35, 44, 28, 32]. For example, TENT [44] optimizes the affine parameters in batch normalization layers by minimizing the entropy of the prediction probability. MEMO [53] applies variant augmentation methods to a single test sample and optimizes model parameters by minimizing the entropy of the prediction probability. A recent advancement in test-time adaptation with distribution shift, which introduces the concept of universal TTA to address domain non-stationarity and temporal correlation, ensuring robust model performance across diverse scenarios [34]. The most relevant of these traditional TTA methods to us is T3A [24], which achieves test-time adaptation by adjusting the trained linear classifier using prototypes. Compared to T3A, which naively stores typical test samples, we achieve continuous adaptation by estimating the distribution of test samples.

**TTA for VLMs**. To enhance the performance of VLMs during testing, TPT [41] introduces adaptive text prompts and optimizes the prompts through entropy minimization. Building on this, DiffTPT [13] leverages pre-trained stable diffusion models to generate diverse augmented data for use in test-time prompt tuning. However, TPT and DiffTPT rely heavily on gradient backpropagation to optimize the prompts, making them computationally expensive and resource-intensive. TDA [25] proposes a lightweight test-time adaption method by storing representative test samples. Building upon TDA, Boostadapter [55] enhances the cache sample selection strategy through Regional Bootstrapping. Compared to TDA and Boostadapter, which naively stores typical test samples, we achieve continuous adaptation by estimating the distribution of test samples, leading to a more adaptive solution.

**Distribution estimation for recognition.** Distribution estimation plays a crucial role in adapting recognition models by leveraging data's statistical properties for dynamic updates. This approach is particularly effective when encountering new classes or shifting data distributions [19, 48]. For example, [3] developed an open-world recognition system that continuously learns new object categories by evolving the nearest class mean algorithm into a nearest non-outlier variant. Similarly, Prototypical Networks [42] utilize distribution estimation by defining class prototypes as the mean of embedded support examples; classification then relies on metric distances, yielding strong performance in few-shot and zero-shot settings. Further addressing dynamic data, previous method [9] proposed a system for continual prototype evolution, facilitating online learning from non-stationary streams via efficient memory management and a novel objective function. Recently, a training-free CLIP adaptation method has been proposed by introducing Gaussian Discriminant Analysis [48], which estimates class means and shared covariance from data to build an ensemble classifier integrating zero-shot CLIP predictions. Building upon these principles, particularly from the continual learning literature, this paper introduces `Dota` to enhance the test-time performance of vision-language models.

**Vision-language models** have demonstrated strong vision understanding capabilities benefiting from training on large-scale datasets [39, 51, 30]. Among them, CLIP [39] is the most representative method by maximizing the similarity between image and their corresponding text embeddings. To further enhance performance of CLIP on downstream tasks, prompt learning-based methods have been proposed by optimizing the prompts of the text encoder [57, 58, 2, 27, 2]. Moreover, to reduce the computational cost associated with gradient calculations in prompt learning, efficient CLIP adaptation methods have been introduced [16, 54, 48, 31, 50]. These methods enable downstream task adaptation using only a small number of training samples in the embedding space. Orthogonal to above methods, this paper focuses on continuously adapting to environments during testing by leveraging test samples.

## 3 Method

### 3.1 Zero-Shot Classification with Prompt

**Zero-shot classification.** During the pre-training stage, CLIP[2] trains its image and text encoders using large-scale image-text pairs. This is achieved by maximizing the cosine similarity between the image

---

[2]`Dota` is also applicable to other similar models.

and text embeddings through contrastive loss. Unlike traditional classifiers trained on closed-set labels, CLIP leverages open-set semantic information in the image-text pairs to learn a broader range of visual concepts. Consequently, during the test stage, CLIP can perform zero-shot classification without additional training. Specifically, given a test sample $\boldsymbol{x}$ for $K$-class classification, where $\boldsymbol{x}$ represents the image embedding obtained from the image encoder, the corresponding zero-shot prediction probability $P_k^{\texttt{zs}}$ for class $k$ is calculated as:

$$P_k^{\texttt{zs}}(y = k|\boldsymbol{x}) = \frac{\exp(\cos(\boldsymbol{x}, \boldsymbol{w}_k)/\tau)}{\sum_{k=1}^K \exp(\cos(\boldsymbol{x}, \boldsymbol{w}_k)/\tau)}, \tag{1}$$

where $\texttt{zs}$ refers to zero-shot. $\boldsymbol{w}_k$ is the classification weight for class $k$, obtained by encoding the corresponding prompt, e.g., "a photo of {class}", with the class token replaced by the specific category name. $\tau$ is the learned temperature parameter in CLIP, and $\cos(\cdot, \cdot)$ denotes the cosine similarity. The above classification process can be understood as comparing the obtained image embedding with the text prompt and selecting the most similar category as the final decision.

## 3.2 Distributional Test-time Adaptation

**Key motivation.** When CLIP is deployed across different environments, its performance often deteriorates due to changes in the data distribution, especially when the test data significantly deviates from the training data. TTA can effectively enable the foundational model to adapt quickly to new environments during the test phase. Current state-of-the-art methods typically maintain a cache of representative samples from various classes, which guide the classification of subsequent test samples [25, 55]. However, due to the limited cache size, cache-based TTA methods face several challenges:

- Limited information storage: These methods store embeddings for only a limited number of samples, which restricts the breadth and depth of information captured. As a result, they may fail to adequately represent the full complexity and nuances of the test data distribution.

- Lack of deep association learning: These methods primarily rely on cached samples for similarity matching or guidance, rather than learning or updating the model's understanding of the deeper, intrinsic relationships between sample features and their corresponding semantic labels in the new test environment.

- Significant risk of test-time forgetting: Due to the above limitations, when the cache must be updated (e.g., by replacing older samples), the model struggles to retain the adaptive knowledge gained previously. This leads to a high risk of catastrophic forgetting, compromising both performance and stability as new data arrives.

To address these challenges, as shown in Fig. 1, we propose distributional test-time adaptation (Dota), which continuously estimates the evolving test sample distribution during testing. By leveraging Bayes' theorem to infer the posterior distribution of different classes, Dota allows for dynamic adaptation based on an accurate understanding of the current data distribution. This distributional estimation ensures richer, more reliable information for classification, overcoming the limitations of cache-based methods and reducing the risk of forgetting, thus maintaining stable performance.

**Classification with classical Gaussian discriminant analysis.** Formally, inspired by classical Gaussian discriminant analysis [19], we assume that the embedding distribution of each class $k$ follows a Gaussian distribution, i.e., $P(\boldsymbol{x}|y=k) = \mathcal{N}(\boldsymbol{\mu}_k, \boldsymbol{\Sigma}_k)$, where $\boldsymbol{\mu}_k$ and $\boldsymbol{\Sigma}_k$ are the mean vector and covariance matrix of class $k$, respectively. Using Bayes' theorem, the posterior probability $P(y=k|\boldsymbol{x})$ of class $k$ can be given by

$$P(y=k|\boldsymbol{x}) = \frac{P(\boldsymbol{x}|y=k)P(y=k)}{P(\boldsymbol{x})},$$

where $P(\boldsymbol{x}) = \sum_{k=1}^K P(\boldsymbol{x}|y=k)P(y=k)$ and $P(y=k)$ is the prior probability. In practice, we set $P(y=k)$ to $1/K$ for simplicity. Then $P(y=k|\boldsymbol{x})$ can be obtained with

$$P(y=k \mid \boldsymbol{x}) = \frac{\exp(f_k(\boldsymbol{x}))}{\sum_{k=1}^K \exp(f_k(\boldsymbol{x}))}, \tag{2}$$

where $f_k(\boldsymbol{x}) = -\frac{1}{2}(\boldsymbol{x}-\boldsymbol{\mu}_k)^T \boldsymbol{\Sigma}_k^{-1}(\boldsymbol{x}-\boldsymbol{\mu}_k) - \frac{1}{2}\log|\boldsymbol{\Sigma}_k|$. The discriminant function $f_k(\boldsymbol{x})$ measures how well a sample $\boldsymbol{x}$ fits the distribution of class $k$. The detail can be found in the Appendix A.6. For

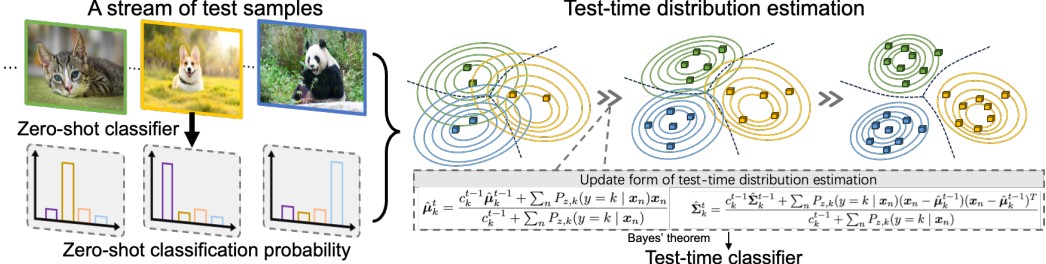

Figure 2: Pipeline of `Dota`. During test time, a stream of test samples is evaluated with original zero-shot classifier, and we estimate the distributions for the test samples during testing, enabling the model to continually learn from the test samples and the zero-shot classification probabilities. *As the number of test samples increases, the estimated test sample data distribution will become more accurate.* Finally the test-time classifier can be obtained with the estimated distributions according to Bayes' theorem for test-time adaptation.

the $k$-th class with $N_k$ training samples, the mean and covariance matrix of different classes can be estimated as follows:

$$\hat{\boldsymbol{\mu}}_k = \frac{\sum_{n=1}^{N_k} \boldsymbol{x}_n}{N_k}, \hat{\boldsymbol{\Sigma}}_k = \frac{\sum_{n=1}^{N_k}(\boldsymbol{x}_n - \hat{\boldsymbol{\mu}}_k)(\boldsymbol{x}_n - \hat{\boldsymbol{\mu}}_k)^T}{N_k}, \tag{3}$$

where $\hat{\boldsymbol{\mu}}_k$ and $\hat{\boldsymbol{\Sigma}}_k$ represent the estimated mean and covariance matrix of the $k$-th class, respectively. $\boldsymbol{x}_n$ denotes the input embeddings. However, there are two problems with the estimation of the mean and covariance in Eq. 3 that make it unsuitable for test-time distribution estimation. First, during testing, class labels are not available to compute the mean and covariance for class $k$. Second, during testing, it is impossible to access all test samples at once for distribution estimation, as the test samples are provided in the form of a data stream. We will address these two issues one by one below.

**Parameter estimation with zero-shot predictive probability.** When conducting test-time distribution estimation, one main challenge is that we cannot access to the ground-truth labels for the $N$ test samples. Therefore, we try to use the zero-shot predictive probability to estimate the distribution [19]. To this end we introduce the Prop. 3.1. The proof can be found in the appendix A.1.

**Proposition 3.1** (Parameter estimation with zero-shot predictive probability using the Expectation–Maximization (EM) algorithm)*. Let $\{P_k^{zs}(y = k \mid \boldsymbol{x}_n)\}_{k=1}^K$ be the zero-shot prediction probabilities for $n$-th test sample. The means $\{\hat{\boldsymbol{\mu}}_k\}_{k=1}^K$ and covariances $\{\hat{\boldsymbol{\Sigma}}_k\}_{k=1}^K$ of the distribution can be estimated as a single iteration of the EM algorithm, where the expectation step computes the zero-shot prediction and the maximization step updates the parameters by maximizing the likelihood. The estimates are:*

$$\hat{\boldsymbol{\mu}}_k = \frac{\sum_{n=1}^N P_k^{zs}(y = k \mid \boldsymbol{x}_n)\boldsymbol{x}_n}{\sum_{n=1}^N P_k^{zs}(y = k \mid \boldsymbol{x}_n)}, \hat{\boldsymbol{\Sigma}}_k = \frac{\sum_{n=1}^N P_k^{zs}(y = k \mid \boldsymbol{x}_n)(\boldsymbol{x}_n - \hat{\boldsymbol{\mu}}_k)(\boldsymbol{x}_n - \hat{\boldsymbol{\mu}}_k)^T}{\sum_{n=1}^N P_k^{zs}(y = k \mid \boldsymbol{x}_n)}. \tag{4}$$

Prop. 3.1 shows that the estimation process with zero-shot predictive probability. The estimation can also be intuitively understood as reweighting, where the zero-shot probabilities are used as weights to adjust the contributions of different samples, thereby mitigating the impact of the potential inaccuracies in the zero-shot prediction.

**Online test-time distribution estimation.** When estimating data distribution at test time, one another challenge is that we evaluate the test samples sequentially in a streaming manner instead of accessing all samples simultaneously. This necessitates a strategy to appropriately adjust the estimation method in Eq. 4 through effective initialization, and then allowing the parameters to be updated quickly as new test samples arrive. To achieve this goal, `Dota` maintains the distribution information of different classes (i.e., mean and covariance matrix) during testing, and updates its distribution information based on its representation information after obtaining new samples. **Initialization of** $\{\hat{\boldsymbol{\mu}}_k, \hat{\boldsymbol{\Sigma}}_k\}_{k=1}^K$. We can initialize the estimated mean of different classes in the following way:

$$\hat{\boldsymbol{\mu}}_k^0 = \omega \mathbf{1} \quad \text{and} \quad \hat{\boldsymbol{\Sigma}}_k^0 = \sigma^2 \boldsymbol{I}, \tag{5}$$

where $\omega$ and $\sigma^2$ is a hyperparameter that determines the initial mean and variance, $\boldsymbol{I}$ is the identity matrix. **Update of** $\{\hat{\boldsymbol{\mu}}_k, \hat{\boldsymbol{\Sigma}}_k\}_{k=1}^K$. We employ the update form described in [8], which is capable of estimating Gaussian distribution parameters in an online setting. Theoretically, for any sequence, the average regret of the update form converges to zero in the limit. Specifically, given a batch of test samples at step $t$, the updated $\hat{\boldsymbol{\mu}}_k^t, \hat{\boldsymbol{\Sigma}}_k^t$ can be computed based on the $\hat{\boldsymbol{\mu}}_k^{t-1}, \hat{\boldsymbol{\Sigma}}_k^{t-1}$ as follows:

$$\hat{\boldsymbol{\mu}}_k^t = \frac{c_k^{t-1}\hat{\boldsymbol{\mu}}_k^{t-1} + \sum P_k^{\texttt{zs}}(y=k \mid \boldsymbol{x}_n)\boldsymbol{x}_n}{c_k^{t-1} + \sum P_k^{\texttt{zs}}(y=k \mid \boldsymbol{x}_n)}, \hat{\boldsymbol{\Sigma}}_k^t = \frac{c_k^{t-1}\hat{\boldsymbol{\Sigma}}_k^{t-1} + \sum P_k^{\texttt{zs}}(y=k \mid \boldsymbol{x}_n)\boldsymbol{S}_k^{t-1}}{c_k^{t-1} + \sum P_k^{\texttt{zs}}(y=k \mid \boldsymbol{x}_n)}, \quad (6)$$

where $\boldsymbol{S}_k^{t-1} = (\boldsymbol{x}_n - \hat{\boldsymbol{\mu}}_k^{t-1})(\boldsymbol{x}_n - \hat{\boldsymbol{\mu}}_k^{t-1})^T$, $c_k^{t-1}$ represents the effective sample size, defined by the cumulative confidences of the observed samples of class $k$ at step $t-1$. Specifically, we set $c_k^0 = 0$ and $c_k^t$ is updated as $c_k^t = c_k^{t-1} + \sum P_k^{\texttt{zs}}(y=k|\boldsymbol{x}_n)$. When we obtain the estimated $\hat{\boldsymbol{\mu}}, \hat{\boldsymbol{\Sigma}}$, we can use Eq. 2 to calculate the test-time adapted posterior probability.

In practice, Eq. 6 is a generalized vector update version that works effectively with different test batch sizes. For consistency with comparison methods, we set the batch size to 1 in our experiments. To reduce computational complexity when inverting the covariance matrix $\hat{\boldsymbol{\Sigma}}_k$, similar to the approach in [1, 14], we approximate the covariance by averaging across all classes, reducing the number of matrix inversions from $K$ to 1, thereby improving efficiency and reducing the number of parameter estimates. Additionally, we apply shrinkage regularization to the precision matrix to enhance the stability of the inversion process as follows: $\hat{\boldsymbol{\Lambda}} = [(1 - \epsilon)\hat{\boldsymbol{\Sigma}} + \epsilon\boldsymbol{I}]^{-1}$, where $\epsilon = 10^{-4}$ is the shrinkage parameter. The term $\epsilon\boldsymbol{I}$ ensures that the eigenvalues of the covariance matrix are well-conditioned, maintaining the desired properties such as positive definiteness and rank stability.

### 3.3   Adaptive fusion of zero-shot and test-time classifier

As the number of test samples increases, the reliability of the estimated test sample distribution improves [8]. However, when the number of test samples is insufficient, the estimated distribution may be unreliable. To address this, we introduce a dynamic zero-shot classification and test-time result fusion approach, allowing the model to rely more on zero-shot classification during stages where the sample size for distribution estimation is insufficient. Formally, the final fusion probability is defined as follows:

$$P_k(y = k|x) = \frac{\exp(\cos(\boldsymbol{x}, \boldsymbol{w}_k)/\tau + \lambda f_k(\boldsymbol{x}))}{\sum_{k=1}^K [\exp(\cos(\boldsymbol{x}, \boldsymbol{w}_k)/\tau + \lambda f_k(\boldsymbol{x}))]}, \quad (7)$$

where $\lambda = \min(\rho c, \eta)$. Here, $c$ represents the number of test samples, and $\rho$ and $\eta$ are hyperparameters that control the weight of the test-time classifier logits. The value of $\lambda$ increases with the number of test samples when this number is insufficient, gradually approaching the maximum value $\eta$. This approach encourages the model to rely on the zero-shot classifier results when the test samples are insufficient to estimate the distribution, mitigating the potential negative impact of the test-time classifier. The whole pseudo code is shown in Alg. 1.

The additive fusion in Eq. 7 is a heuristic yet effective strategy designed to balance the robust prior knowledge from the pre-trained model with dynamic adaptation to test-time shifts. This logit-level combination is computationally efficient and has been widely adopted as a common and effective practice in the TTA literature for leveraging pre-trained knowledge while incorporating new test-time information[25, 52]. While this approach is largely heuristic, it dynamically ensures minimal interference in early stages while effectively refining predictions as Dota's per-class modeling becomes more accurate, proving highly effective in practice.

## 4   Experiments

**Benchmarks.** Consistent with prior works [41, 13, 25], we conduct our main experiments on cross-domain generalization and natural distribution shifts (NDS) scenarios. In the cross-domain generalization scenario, we evaluate the performance of the model across 10 diverse image classification datasets, each representing a distinct domain with different classes: Aircraft [33], Caltech101 [12], Cars [29], DTD [7], EuroSAT [20], Flower102 [36], Food101 [4], Pets [37], SUN397 [49], and UCF101 [43]. This benchmark provides a comprehensive evaluation of the adaptability of the

**Algorithm 1:** The pseudocode of `Dota`.

---

**Input:** The embedding of $N$ test samples $\{\boldsymbol{x}_n\}_{n=1}^{N}$ in an streaming way, zero-shot classification weights $[\boldsymbol{w}_1, \cdots, \boldsymbol{w}_K]$;

Initializing the distribution of different class;

**for** each test sample $\boldsymbol{x}_i$ **do**

    Obtain the zero-shot probability with Eq. 1;

    Update the distribution of different class with Eq. 6;

    Obtain the test-time classification probability with Eq. 2;

    Obtain the final classification result with Eq. 7.

**end for**

---

Table 1: Top-1 accuracy (%) under the cross-domain generalization scenario. "PE" represents the use of prompt enhancement[38].

| Method | Aircraft | Caltech101 | Cars | DTD | EuroSAT | Flower102 | Food101 | Pets | SUN397 | UCF101 | Average |
|---|---|---|---|---|---|---|---|---|---|---|---|
| Zero-Shot | 23.22 | 93.55 | 66.11 | 45.04 | 50.42 | 66.99 | 82.86 | 86.92 | 65.63 | 65.16 | 64.59 |
| TPT[41] | 24.78 | 94.16 | 66.87 | 47.75 | 42.44 | 68.98 | 84.67 | 87.79 | 65.50 | 68.04 | 65.10 |
| DiffTPT[13] | 25.60 | 92.49 | 67.01 | 47.00 | 43.13 | 70.10 | 87.23 | 88.22 | 65.74 | 62.67 | 65.47 |
| TDA[25] | 23.91 | 94.24 | 67.28 | 47.40 | 58.00 | 71.42 | 86.14 | 88.63 | 67.62 | 70.66 | 67.53 |
| BoostAdapter[55] | 27.45 | 94.77 | 69.30 | 45.69 | 61.22 | 71.66 | 87.17 | 89.51 | 68.09 | 71.93 | 68.68 |
| HisTPT[52] | 26.90 | 94.50 | 69.20 | 48.90 | 49.70 | 71.20 | 89.30 | 89.10 | 67.20 | 67.60 | 67.60 |
| ZERO[11] | 25.21 | 93.66 | 68.04 | 46.12 | 34.33 | 67.68 | 86.53 | 87.75 | 65.03 | 67.77 | 64.21 |
| Dota | 26.25 | 94.16 | 69.56 | 47.64 | 62.78 | 75.23 | 87.08 | 92.01 | 69.80 | 72.54 | 69.71 |
| DMN w/PE [56] | 30.03 | 95.38 | 67.96 | 55.85 | 59.43 | 74.49 | 85.08 | 92.04 | 70.18 | 72.51 | 70.30 |
| Dota w/ PE | 29.82 | 94.85 | 69.06 | 55.97 | 58.35 | 77.06 | 87.07 | 92.40 | 70.97 | 74.86 | 71.04 |

model during test time across various class spaces. For the natural distribution shifts scenario, we utilize multiple datasets including ImageNet [10], ImageNet-A [22], ImageNet-R [21], ImageNet-S [46] and ImageNet-V2 [40], which serve as measures of the robustness of our approach. We also evaluate `Dota` using medical image datasets and a pathology-pretrained foundation model [23], where the labels were converted into descriptive sentences for evaluation (e.g., transforming "tumor" into "H&E image of a tumor"). The evaluation includes three datasets: the Kather Colon dataset [26], comprising nine different tissue types; the PanNuke dataset [15], focusing on benign versus malignant classifications; and the WSSS4LUAD dataset [18], which distinguishes between tumor and normal samples.

**Comparison Method.** We compare the proposed method with the following approaches: (1) CLIP's zero-shot method, which utilizes an ensemble of 80 prompts [39]; (2) Prompt-based training methods, including TPT [41], DiffTPT [13], and Historical Prompt Tuning [52]. These methods focus on optimizing input prompts rather than modifying the model's parameters. TPT and DiffTPT adapt the prompt at test time, with DiffTPT also introducing more diverse test sample augmentation using a diffusion model. Historical Prompt Tuning uses prior task knowledge to refine prompts for better task adaptation. (3) Efficient test-time adaptation methods, which do not require backpropagation and rely on a cache of representative samples for adaptation, as demonstrated in [11, 25, 56, 55]. The results for the above methods are obtained from the original papers.

Table 2: Performance under the NDS scenario.

| Method | ImageNet | ImageNet-A | ImageNet-R | ImageNet-S | Average |
|---|---|---|---|---|---|
| Zero-Shot | 68.34 | 49.89 | 77.65 | 48.24 | 61.03 |
| TPT [41] | 68.98 | 54.77 | 77.06 | 47.94 | 62.19 |
| DiffTPT [13] | 70.30 | 55.68 | 75.00 | 46.80 | 61.95 |
| TDA[25] | 69.51 | 60.11 | 80.24 | 50.54 | 65.10 |
| ZERO [11] | 69.31 | 59.61 | 77.22 | 48.40 | 63.64 |
| Dota | 70.69 | 61.50 | 81.21 | 51.84 | 66.31 |

Table 3: Performance comparison across datasets on PLIP model.

| Dataset | Zero-Shot | TDA | CTA |
|---|---|---|---|
| Kather | 45.60 | 49.35 | 61.92 |
| PanNuke | 69.49 | 69.70 | 74.68 |
| WSSS4LUAD | 70.31 | 71.96 | 75.07 |
| average | 61.80 | 63.67 | 70.56 |

## 4.1 Comparison with state-of-the-arts methods

**Results under the cross-domain generalization scenario.** We first compare `Dota` with state-of-the-art methods under the cross-domain generalization scenario across 10 diverse image classification datasets, each from a distinct domain with different classes. Tab. 1 presents the experimental results. The proposed method achieved the best performance on most datasets and the top two performance on all datasets. For example, when using the ViT-B/16 backbone network, the average performance was improved by 1.03%.

**Results under the natural distribution shifts scenario.** We then compare `Dota` with state-of-the-art methods in the context of natural distribution shifts. Tab. 2 presents the experimental results, revealing the following key observation. Leveraging distribution modeling of the representation of test data, `Dota` achieves superior performance without requiring gradient backpropagation. For instance, using the CLIP-ViT-B/16 backbone network, `Dota` outperforms the second-best method by an average of 1.21%, achieving state-of-the-art results across all datasets.

**Performance validation on pathological image classification with PLIP.** To validate the performance of the proposed method across other pretrained models and application scenarios, we conducted experiments using the PLIP model in the context of pathological image classification. The results are presented in Tab. 3. As shown in the table, the proposed method achieved superior performance, with an average improvement of 6.89% over TDA.

Table 4: Comparisons of our `Dota` with other methods in terms of efficiency (*Testing Time*) and effectiveness (*Accuracy*).

| Method | Testing Time | Accuracy | Gain |
|---|---|---|---|
| Zero-Shot | 11.82min | 68.34 | 0 |
| TPT | 447min | 68.98 | +0.64 |
| DiffTPT | 1346min | 70.30 | +1.96 |
| TDA | **22min** | 69.51 | +1.17 |
| `Dota` (Ours) | **22min** | **70.69** | **+2.35** |

Table 5: Comparisons of our `Dota` with other methods on the ImageNetV2 dataset, where each class contains only 10 samples.

| Method | ViT-B/16 | ResNet-50 |
|---|---|---|
| Zero-Shot | 61.88 | 52.91 |
| TDA | 64.67 | 55.54 |
| `Dota` (All test samples) | 64.50 | 55.19 |
| `Dota` (last 50% test samples ) | 65.20 | 55.72 |

**Inference time comparison.** To illustrate the efficiency of the proposed method, we conduct evaluation about the inference time using the ViT-B/16 backbone on the ImageNet [10] dataset. The experimental results are shown in Tab. 4. From the table, we can see that the proposed method is faster than the methods that require gradient backpropagation. For example, `Dota` is 24 times faster than TPT, and 61 times faster than DiffTPT. Therefore, test-time adaptation methods that require gradient backpropagation may not be applicable during deployment due to the performance limitations of the inference device. At the same time, compared with TDA, the speed of the proposed method is comparable, but the performance is higher.

**Failure case study.** While our approach highlights the advantages of continuously estimating the distribution of test data and adapting to it, it does not consistently outperform TDA across all datasets, particularly those with a limited number of samples. For instance, as shown in Tab. 5, on the ImagenetV2 [40], which contains only 10 samples per class, the performance of `Dota` is slightly lower than TDA. This is likely because the limited number of samples per class is insufficient to accurately estimate the data distribution online. However, its performance shows a marked improvement on the latter 50% of the test samples. This observation suggests that the proposed model has the potential for further enhancement as the number of available test samples increases.

Table 6: Performance of `Dota` and TDA, comparing overall accuracy and the last 50% of test samples to show continuous adaptability. The results show that the performance of our method is continual improving. However, TDA is different, and its performance has declined on several datasets.

| Method | Aircraft | Caltech101 | Cars | DTD | Flower102 | Food101 | Pets | SUN397 | UCF101 |
|---|---|---|---|---|---|---|---|---|---|
| TDA (all test samples) | 23.91 | 94.24 | 67.28 | 47.40 | 71.42 | 86.14 | 88.63 | 67.62 | 70.66 |
| TDA (last 50% test samples) | 26.57 | 93.59 | 66.95 | 46.22 | 71.75 | 86.02 | 89.26 | 67.86 | 72.20 |
| `Dota` (All test samples) | 26.25 | 94.16 | 69.56 | 47.64 | 75.23 | 87.08 | 92.01 | 69.80 | 72.54 |
| `Dota` (last 50% test samples) | 27.65 | 94.65 | 69.98 | 50.47 | 76.46 | 87.12 | 93.30 | 70.69 | 73.73 |

## 4.2 Ablation studies and further analysis

**Analysis of continuous learning ability and test-time forgetting of TDA.** We conducted this experiment to validate our research motivation and demonstrate experimentally that `Dota` possesses the capability for continuous learning. During testing on the ImageNet dataset, we recorded the performance of the most recent 5,000 test samples and compared it with the original zero-shot classifier's performance. We analyzed the relationship between the improvement in model performance and the number of test samples processed. The results are illustrated in Fig. 3. From the experimental results, it can be shown that the proposed method progressively enhances model performance as the number of test samples increases. In contrast, TDA shows an initial improvement that subsequently declines, indicating its inability to con-

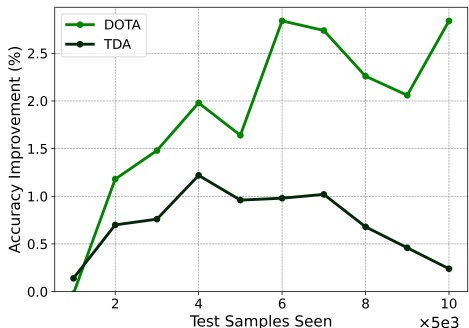

Figure 3: Improvement of different methods in model performance as the number of encountered test samples increases.

tinuously learn from the test data stream. Further analysis on additional datasets, shown in Tab. 6, highlights the performance on the last 50% of test samples as well as all samples. The results clearly demonstrate that the performance of the last 50% of test samples for `Dota` is significantly higher than the overall performance. This improvement can be attributed to the increasing reliability of the estimated distribution as more test samples are observed. However, TDA exhibits a different pattern, with its performance declining on several datasets.

Table 7: Ablation study to compare the performance of `Dota` with two variants: (1) rely solely on high-confidence samples and their predictive labels for estimating the distribution; (2) using only the mean, without updating the covariance matrix in the estimation of the Gaussian distribution.

| Method | Aircraft | Caltech101 | Cars | DTD | EuroSAT | Flower102 | Food101 | Pets | SUN397 | UCF101 | Average |
|---|---|---|---|---|---|---|---|---|---|---|---|
| `Dota` | 26.25 | 94.16 | 69.56 | 47.64 | 62.78 | 75.23 | 87.08 | 92.01 | 69.80 | 72.54 | 69.71 |
| high-confidence samples only | 24.63 | 94.12 | 67.11 | 46.34 | 53.28 | 72.07 | 86.47 | 90.68 | 68.24 | 69.28 | 67.22 |
| | -1.62 | -0.04 | -2.45 | -1.30 | -9.50 | -3.16 | -0.61 | -1.33 | -1.56 | -3.26 | -2.49 |
| w/o covariance | 25.29 | 94.16 | 67.47 | 45.62 | 55.06 | 71.34 | 86.44 | 90.57 | 67.88 | 69.34 | 67.32 |
| | -0.96 | 0.00 | -2.09 | -2.02 | -7.72 | -3.89 | -0.64 | -1.44 | -1.92 | -3.20 | -2.39 |

**The Importance of distribution estimation using predictive probabilities of all test samples in an EM framework.** We compared the performance of the `Dota` with a simplified version that only uses high-confidence samples and their corresponding predictive labels for estimating the distribution. The experimental results are shown in Tab. 7. The experimental results demonstrate that updating the data distribution using the proposed zero-shot predictive probabilities achieves better performance compared to using pseudo-labels from only high-confidence samples. For instance, overall performance sees a decline of approximately 2.49% when relying solely on high-confidence pseudo-label.

**The necessity of distribution estimation.** We compared the performance of the `Dota` with a simplified version that uses only the mean, excluding the estimation of the Gaussian distribution by removing the updates to the covariance matrices. This experiment aimed to understand the necessity of continual distribution estimation in enhancing model accuracy. The experimental results are shown in Tab. 7. The fifth row in the table presents the accuracy reductions across different datasets when the covariance matrix is not updated. The results indicate a consistent decrease in accuracy across all datasets, with a particularly notable drop of 7.72% on the EuroSAT dataset. These findings highlight the importance of continual distribution estimation.

Table 8: Hyperparameters analysis on the $\sigma^2$ and $(\rho, \eta)$ combinations.

| $\sigma^2$ | 0.0001 | 0.001 | 0.002 | 0.004 | 0.008 | 0.02 |
|---|---|---|---|---|---|---|
| Acc | 70.72 | 70.72 | 70.69 | 70.70 | 70.60 | 70.42 |

| $\eta \backslash \rho$ | 0.005 | 0.01 | 0.02 | 0.03 |
|---|---|---|---|---|
| 0.2 | 70.69 | 70.66 | 70.59 | 70.54 |
| 0.3 | 70.64 | 70.55 | 70.36 | 70.28 |
| 0.4 | 70.64 | 70.51 | 70.24 | 70.13 |
| 0.5 | 70.64 | 70.48 | 70.15 | 70.00 |

**Hyperparameters analysis.** To validate the sensitivity of our model to hyperparameters, we conduct systematic experiments and analyses. First, we evaluate the hyperparameter $\sigma^2$ while keeping other fixed. The results showed minimal impact on accuracy, with performance ranging from 70.42 to 70.72.

Next, we test different $\rho$ and $\eta$ combinations, observing stable performance across combinations. For instance, accuracy ranged from 70.69 to 70.00 as $\rho$ and $\eta$ varied. Another hyperparameter $\omega$ we consistently set to 0.001. Notably, all hyperparameter combinations show that the proposed method outperforms the original zero-shot classifier, indicating that TTA can usually significantly enhance performance even without a validation set for hyperparameter tuning.

**The necessity of adaptive fusion of zero-shot and test-time classifier**. We conduct ablation study to show that adaptive fusion of zero-shot and test-time classifier is necessary. The specific experimental results are shown on the Tab. 8. It can be observed that as $\rho$ increases (indicating the diminishing effect of dynamic fusion), the performance of `Dota` consistently decreases.

## 5 Conclusion and Limitations

In this paper, we proposed DistributiOnal Test-time Adaptation (`Dota`), a method that overcomes the limitations of cache-based test-time adaptation by continuously estimating the underlying distribution of test data. Unlike traditional methods, which are constrained by fixed cache sizes and suffer from catastrophic forgetting, `Dota` dynamically updates class distributions, leading to more efficient and adaptive test-time inference. Our experiments show that `Dota` significantly reduces forgetting, improves performance, and offers over 20 times faster inference compared to test-time prompt training methods. This approach provides an effective solution for deploying vision-language models in dynamic environments, offering a scalable and computationally efficient alternative to traditional adaptation techniques.

This method has three main limitations that suggest directions for future work. First, its single Gaussian distribution assumption restricts its ability to model complex data, which could be addressed by exploring more flexible models. Second, the assumption of uniform class priors is inconsistent with real-world class imbalances and could be improved with an adaptive prior estimation mechanism. Finally, the heuristic fusion strategy could be replaced by a more principled approach that dynamically adjusts weights based on real-time sample uncertainty.

## Acknowledgments and Disclosure of Funding

This work is supported by the National Natural Science Foundation of China (No. 624B2100, No. 62376193, No. 61925602, No. T2522008, and No. 62272055), New Cornerstone Science Foundation through the XPLORER PRIZE. We thank Prof. Qinghua Hu for his help with the manuscript. Mike Shou does not receive any funding for this work. The authors appreciate the valuable feedback from anonymous reviewers.

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

# A Appendix

## A.1 Proof of Proposition 3.1

The EM algorithm is an iterative method used for finding maximum likelihood estimates of parameters in statistical models with latent (unobserved) variables. It consists of two main steps:

(1) Expectation Step (E-step): Compute the expected value of the log-likelihood function, with respect to the current estimate of the distribution of the latent variables.

(2) Maximization Step (M-step): Maximize this expected log-likelihood to update the parameter estimates.

In the context of Gaussian Discriminant Analysis (GDA) for classification, the latent variables correspond to the true class labels of the data points, which are unobserved during testing.

**Assumption A.1** (Class-Conditional Distributions). For each class $k \in \{1, 2, \ldots, K\}$, the data embedding $\boldsymbol{x}$ follows a Gaussian distribution:

$$P(\boldsymbol{x} \mid y = k) = \mathcal{N}(\boldsymbol{\mu}_k, \boldsymbol{\Sigma}_k)$$

where $\boldsymbol{\mu}_k$ and $\boldsymbol{\Sigma}_k$ are the mean vector and covariance matrix of class $k$, respectively.

**Assumption A.2** (Prior Probabilities). The prior probability for each class is uniform: $P(y = k) = \frac{1}{K}$.

The objective is how to estimate the parameters $\{\boldsymbol{\mu}_k, \boldsymbol{\Sigma}_k\}_{k=1}^{K}$ using the EM algorithm, leveraging the zero-shot predictive probabilities $P_k^{\text{zs}}(y = k \mid \boldsymbol{x}_n)$ as part of the expectation step.

**E-Step: Compute Expected Log-Likelihood.** In the E-step, we compute the expectation of the log-likelihood with respect to the posterior distribution of the latent variables given the current parameter estimates. Given the zero-shot predictive probabilities $P_k^{\text{zs}}(y = k \mid \boldsymbol{x}_n)$, we treat them as the responsibilities (i.e., the posterior probabilities) in the E-step:

$$\pi_{nk} = P_k^{\text{zs}}(y = k \mid \boldsymbol{x}_n)$$

The expected log-likelihood $Q$ is then:

$$Q(\{\boldsymbol{\mu}_k, \boldsymbol{\Sigma}_k\} \mid \text{current estimates}) = \sum_{n=1}^{N} \sum_{k=1}^{K} \pi_{nk} \left[ \log \mathcal{N}(\boldsymbol{x}_n \mid \boldsymbol{\mu}_k, \boldsymbol{\Sigma}_k) + \log \frac{1}{K} \right]$$

**M-Step: Maximize the Expected Log-Likelihood.** In the M-step, we maximize $Q$ with respect to $\{\boldsymbol{\mu}_k, \boldsymbol{\Sigma}_k\}_{k=1}^{K}$. **Maximizing with Respect to $\boldsymbol{\mu}_k$.** To find the optimal $\boldsymbol{\mu}_k$, take the derivative of $Q$ with respect to $\boldsymbol{\mu}_k$ and set it to zero. First, expand the Gaussian log-probability:

$$\log \mathcal{N}(\boldsymbol{x}_n \mid \boldsymbol{\mu}_k, \boldsymbol{\Sigma}_k) = -\frac{1}{2}(\boldsymbol{x}_n - \boldsymbol{\mu}_k)^{\top} \boldsymbol{\Sigma}_k^{-1}(\boldsymbol{x}_n - \boldsymbol{\mu}_k) - \frac{1}{2} \log |\boldsymbol{\Sigma}_k| - \frac{d}{2} \log(2\pi)$$

where $d$ is the dimensionality of $\boldsymbol{x}_n$. Focusing on terms involving $\boldsymbol{\mu}_k$:

$$Q_\mu = -\frac{1}{2} \sum_{n=1}^{N} \pi_{nk}(\boldsymbol{x}_n - \boldsymbol{\mu}_k)^{\top} \boldsymbol{\Sigma}_k^{-1}(\boldsymbol{x}_n - \boldsymbol{\mu}_k)$$

Take the derivative with respect to $\boldsymbol{\mu}_k$ and set to zero:

$$\frac{\partial Q_\mu}{\partial \boldsymbol{\mu}_k} = \sum_{n=1}^{N} \pi_{nk} \boldsymbol{\Sigma}_k^{-1}(\boldsymbol{x}_n - \boldsymbol{\mu}_k) = 0$$

Solving for $\boldsymbol{\mu}_k$:

$$\sum_{n=1}^{N} \pi_{nk} \boldsymbol{x}_n = \left( \sum_{n=1}^{N} \pi_{nk} \right) \boldsymbol{\mu}_k$$

$$\boldsymbol{\mu}_k = \frac{\sum_{n=1}^{N} \pi_{nk} \boldsymbol{x}_n}{\sum_{n=1}^{N} \pi_{nk}}$$

This corresponds directly to the provided update formula for $\hat{\boldsymbol{\mu}}_k$:

$$\hat{\boldsymbol{\mu}}_k = \frac{\sum_{n=1}^{N} P_k^{\mathtt{zs}}(y = k \mid \boldsymbol{x}_n)\boldsymbol{x}_n}{\sum_{n=1}^{N} P_k^{\mathtt{zs}}(y = k \mid \boldsymbol{x}_n)}$$

**Maximizing with Respect to $\boldsymbol{\Sigma}_k$.** Similarly, to find the optimal $\boldsymbol{\Sigma}_k$, consider the terms in $Q$ involving $\boldsymbol{\Sigma}_k$:

$$Q_\Sigma = -\frac{1}{2}\sum_{n=1}^{N} \pi_{nk}\left[(\boldsymbol{x}_n - \boldsymbol{\mu}_k)^\top \boldsymbol{\Sigma}_k^{-1}(\boldsymbol{x}_n - \boldsymbol{\mu}_k) + \log|\boldsymbol{\Sigma}_k|\right]$$

Take the derivative with respect to $\boldsymbol{\Sigma}_k^{-1}$ (using matrix derivative identities) and set to zero:

$$\frac{\partial Q_\Sigma}{\partial \boldsymbol{\Sigma}_k^{-1}} = \frac{1}{2}\sum_{n=1}^{N} \pi_{nk}\left[(\boldsymbol{x}_n - \boldsymbol{\mu}_k)(\boldsymbol{x}_n - \boldsymbol{\mu}_k)^\top - \boldsymbol{\Sigma}_k\right] = 0$$

Solving for $\boldsymbol{\Sigma}_k$:

$$\boldsymbol{\Sigma}_k = \frac{\sum_{n=1}^{N} \pi_{nk}(\boldsymbol{x}_n - \boldsymbol{\mu}_k)(\boldsymbol{x}_n - \boldsymbol{\mu}_k)^\top}{\sum_{n=1}^{N} \pi_{nk}}$$

This aligns with the provided update formula for $\hat{\boldsymbol{\Sigma}}_k$:

$$\hat{\boldsymbol{\Sigma}}_k = \frac{\sum_{n=1}^{N} P_k^{\mathtt{zs}}(y = k \mid \boldsymbol{x}_n)(\boldsymbol{x}_n - \hat{\boldsymbol{\mu}}_k)(\boldsymbol{x}_n - \hat{\boldsymbol{\mu}}_k)^\top}{\sum_{n=1}^{N} P_k^{\mathtt{zs}}(y = k \mid \boldsymbol{x}_n)}$$

Finally, the provided update equation is:

$$\hat{\boldsymbol{\mu}}_k = \frac{\sum_{n=1}^{N} P_k^{\mathtt{zs}}(y = k \mid \boldsymbol{x}_n)\boldsymbol{x}_n}{\sum_{n=1}^{N} P_k^{\mathtt{zs}}(y = k \mid \boldsymbol{x}_n)}, \quad \hat{\boldsymbol{\Sigma}}_k = \frac{\sum_{n=1}^{N} P_k^{\mathtt{zs}}(y = k \mid \boldsymbol{x}_n)(\boldsymbol{x}_n - \hat{\boldsymbol{\mu}}_k)(\boldsymbol{x}_n - \hat{\boldsymbol{\mu}}_k)^\top}{\sum_{n=1}^{N} P_k^{\mathtt{zs}}(y = k \mid \boldsymbol{x}_n)}$$

are precisely the M-step updates obtained by maximizing the expected log-likelihood in the EM algorithm, where the E-step uses the zero-shot predictive probabilities as responsibilities. This demonstrates that the parameter estimation process described is equivalent to performing a single EM iteration. This single EM iteration leverages the zero-shot predictions to adjust the Gaussian parameters, effectively "reweighting" the contributions of different samples based on their inferred class probabilities.

## A.2   Justification for the Gaussian Assumption

Our assumption that the embedding distribution of each class $k$ follows a Gaussian distribution, i.e., $P(x|y = k) = \mathcal{N}(\mu_k, \Sigma_k)$, is grounded in both theoretical principles and empirical evidence.

- Theoretical support: The Central Limit Theorem (CLT) provides a general justification. A high-dimensional embedding can be viewed as an aggregation of numerous lower- and mid-level features. The CLT posits that the sum of many weakly correlated random variables will tend toward a normal distribution, making it reasonable to expect that the composite feature for a single class will approach a multivariate Gaussian distribution.

- Empirical validation: Our own ablation study validates the importance of this assumption; omitting covariance updates, which relies on the Gaussian model, significantly degrades performance (e.g., a 7.72% drop on EuroSAT), confirming the practical utility of Gaussian modeling.

## A.3   Analysis of Catastrophic Forgetting

A key motivation for our work is to mitigate the catastrophic forgetting inherent in cache-based methods like TDA. In the context of TTA, forgetting manifests at a "sample-level"s: due to a fixed-size cache, adapting to new samples necessitates discarding information from older ones. To rigorously validate this issue, we conducted two targeted experiments. The core idea is to have the model, after

a period of continuous adaptation, revisit and re-evaluate its performance on historical test samples. Our experiments revealed two levels of forgetting.

**Forgetting without Significant Distribution Change.** To assess this phenomenon, we designed a two-stage experiment. This process first involves adapting a model on ImageNet and recording its performance (Stage 1), after which we freeze the model's parameters and re-evaluate it on the early 25,000 test samples in the dataset (Stage 2). We hypothesize that if no forgetting occurs, this initial adaptation stage should directly lead to an increase in the model's performance. Conversely, a performance decrease between stage 1 and stage 2 would suggest that the model has experienced forgetting. Experimental results show that TDA suffers from obvious performance degradation, while our method does not.

Table 9: Performance comparison in a two-stage experiment.

| Samples Seen | 10000 | 20000 | 25000 |
|---|---|---|---|
| TDA Stage 1 | 0.42 | 0.71 | 0.76 |
| TDA Stage 2 (Frozen) | 0.17 | 0.57 | 0.29 |
| DOTA Stage 1 | 0.58 | 1.16 | 1.25 |
| DOTA Stage 2 (Frozen) | 2.34 | 2.30 | 2.25 |

**Domain-Level Forgetting.** We challenged the model by adapting it first to an original domain (ImageNet), then to a new, shifted domain (ImageNet-C), and finally re-evaluating its performance on the original domain. TDA's performance failed to improve upon returning to ImageNet, indicating it had forgotten the original domain's features. DOTA, however, showed improvement, demonstrating it retained this knowledge.

Table 10: Domain-level forgetting experiment.

| Method | ImageNet | ImageNet-C brightness | ImageNet (Frozen model) |
|---|---|---|---|
| Zero-Shot | 68.34 | 56.98 | 68.34 |
| TDA | 69.55 | 58.22 | 69.35 |
| DOTA | 70.76 | 60.64 | 70.93 |

We reinterpreted Fig. 3 and added additional experiments. We found that catastrophic forgetting degrades a model's generalization capabilities by eroding previously learned information. More strikingly, we found that under certain conditions, continued adaptation is actively counterproductive, yielding worse results than if adaption had stopped. This is because as we continue to adapt, we forget more than we learn. The Tab. 11 illustrates this clearly. The standard TDA model's performance peaked after 20,000 samples and then degraded. A model whose training was frozen at that point (TDA frozen) consistently outperformed the continuously trained model thereafter. Our proposed method, DOTA, effectively mitigates this degradation and delivers robustly superior performance.

Table 11: Performance degradation during continuous adaptation.

| Samples Seen | 5k | 10k | 15k | 20k | 25k | 30k | 35k | 40k | 45k | 50k | Ave |
|---|---|---|---|---|---|---|---|---|---|---|---|
| TDA | 0.14 | 0.70 | 0.76 | 1.22 | 0.96 | 0.98 | 1.02 | 0.68 | 0.46 | 0.24 | 0.72 |
| TDA (frozen after 20k) | 0.14 | 0.70 | 0.76 | 1.22 | 1.00 | 1.54 | 1.04 | 0.82 | 0.66 | 0.68 | 0.86 |
| DOTA | -0.02 | 1.18 | 1.48 | 1.98 | 1.64 | 2.84 | 2.74 | 2.26 | 2.06 | 2.84 | 1.90 |

### A.4 Results across multiple data ordering

We conducted experiments on five test data in different ordering on multiple datasets. The experimental results are shown in the Tab. 12. From the experimental results, we can see that the order of test data has little effect on the prediction performance.

Table 12: Experimental results across multiple datasets and test data orderings.

| Dataset | Method\Data ordering | 1 | 2 | 3 | 4 | 5 | Average |
|---|---|---|---|---|---|---|---|
| ImageNet | Dota | 70.54 | 70.66 | 70.72 | 70.67 | 70.63 | 70.64 |
| eurosat | Dota | 62.74 | 62.56 | 63.01 | 62.46 | 63.28 | 62.81 |
| OxfordPets | Dota | 92.04 | 91.93 | 92.29 | 92.04 | 92.12 | 92.08 |

## A.5 Performance on non-i.i.d. data streams

We conducted additional experiments to evaluate the model's performance under non-i.i.d. data distribution during testing, using the ImageNet dataset as a benchmark. By employing a Dirichlet distribution, we simulated varying degrees of non-i.i.d. data streams, adjusting the concentration parameter and dividing the dataset into 5 and 10 time slices for analysis. The details of the experiments are shown as follows:

**Time Slices**: We divided the ImageNet dataset into 5 and 10 time slices, where each slice contains varying numbers of samples and class distributions.

**Concentration Parameter** ($[\alpha]_K$): The concentration parameter of the Dirichlet distribution controls the uniformity of class distributions across slices. Smaller $\alpha$ values (e.g., 0.1) create highly uneven distributions, while larger values (e.g., 0.5 and 1) result in more uniform distributions.

**Evaluation Setting**: Since the sizes of sub-datasets for each time slice are unequal, the final average accuracy is a weighted average based on the number of samples in each slice. The experimental results are summarized below.

Table 13: Performance on non-i.i.d. data streams (5 slices).

| $\alpha$ | Slice 1 | Slice 2 | Slice 3 | Slice 4 | Slice 5 | Average |
|---|---|---|---|---|---|---|
| 0.1 | 68.18 | 70.06 | 71.60 | 71.09 | 70.91 | 70.41 |
| 0.5 | 69.55 | 70.78 | 69.43 | 71.95 | 70.92 | 70.58 |
| 1 | 69.41 | 71.64 | 71.05 | 70.01 | 71.81 | 70.83 |

Table 14: Performance on non-i.i.d. data streams (10 slices).

| $\alpha$ | Slice 1 | Slice 2 | Slice 3 | Slice 4 | Slice 5 | Slice 6 | Slice 7 | Slice 8 | Slice 9 | Slice 10 | Average |
|---|---|---|---|---|---|---|---|---|---|---|---|
| 0.1 | 69.99 | 69.31 | 67.34 | 70.23 | 71.49 | 68.8 | 73.13 | 70.35 | 69.89 | 72.39 | 70.39 |
| 0.5 | 68.74 | 69.32 | 72.17 | 71.27 | 69.97 | 69.8 | 70.3 | 70.78 | 72.48 | 70.78 | 70.61 |
| 1 | 67.33 | 70.29 | 70.92 | 68.43 | 69.85 | 71.14 | 72.03 | 72.52 | 71.29 | 71.66 | 70.68 |

From the experimental results, we can see that the model shows strong robustness to non-i.i.d. data streams, with only minimal accuracy decline under small $\alpha$ (e.g., $\alpha = 0.1$).

## A.6 More details and explanation about the $f_k(x)$ in Eq. 2.

The function $f_k(\boldsymbol{x})$, often referred to as the *discriminant function*, measures how well a data point $\boldsymbol{x}$ fits the distribution of class $k$. It is derived from Gaussian Discriminant Analysis and consists of two main components. The first component is the Mahalanobis distance, $-\frac{1}{2}(\boldsymbol{x} - \mu_k)^T \Sigma_k^{-1} (\boldsymbol{x} - \mu_k)$, which calculates the squared distance between $\boldsymbol{x}$ and the class mean $\mu_k$, scaled by the inverse of the covariance matrix $\Sigma_k$. This term captures the similarity of $\boldsymbol{x}$ to the center of the class, considering feature correlations. The second component is the normalization term, $-\frac{1}{2}\log|\Sigma_k|$, which accounts for the determinant of the covariance matrix $\Sigma_k$ and reflects the spread (or volume) of the Gaussian distribution for class $k$. This ensures that classes with larger variances are normalized appropriately. Intuitively, a larger value of $f_k(\boldsymbol{x})$ indicates a higher likelihood that $\boldsymbol{x}$ belongs to class $k$. In classification, $f_k(\boldsymbol{x})$ is used within the softmax function to compute the posterior probability $P(y = k \mid \boldsymbol{x})$, which determines the most likely class for $\boldsymbol{x}$:

$$P(y = k \mid \boldsymbol{x}) = \frac{\exp(f_k(\boldsymbol{x}))}{\sum_{k=1}^{K} \exp(f_k(\boldsymbol{x}))}$$

.

## A.7 CLIP's Superior Calibration Performance

When estimating the distribution, we use zero-shot probabilities, which we argue here to be relatively well-calibrated. Specifically, we provide comparative data on the Expected Calibration Error (ECE) metric to substantiate CLIP's superior calibration performance. We draw on data from [47] and [17] to present the following comparative table of ECE metrics on the ImageNet dataset for different models.

Table 15: Datasets details.

| Dataset | Classes | Validation Size | Test Size | Task |
|---|---|---|---|---|
| ImageNet | 1,000 | N/A | 50,000 | Classification |
| ImageNet-V2 | 1,000 | N/A | 10,000 | Generalization |
| ImageNet-S | 1,000 | N/A | 50,000 | Generalization |
| ImageNet-A | 200 | N/A | 7,500 | Generalization |
| ImageNet-R | 200 | N/A | 30,000 | Generalization |
| Aircraft | 100 | 3,333 | 3,333 | Aircraft recognition |
| Caltech101 | 100 | 1,649 | 2,465 | Object recognition |
| Cars | 196 | 1,635 | 8,041 | Car recognition |
| DTD | 47 | 1,128 | 1,692 | Texture classification |
| EuroSAT | 10 | 5,400 | 8,100 | Remote sensing classification |
| Flowers102 | 102 | 1,633 | 2,463 | Flower recognition |
| Food101 | 101 | 20,200 | 30,300 | Food classification |
| Pets | 37 | 736 | 3,669 | Pet classification |
| SUN397 | 397 | 3,970 | 19,850 | Scene recognition |
| UCF101 | 101 | 1,898 | 3,783 | Action recognition |
| Kather | 2 | 10,718 | 32,154 | Colon classification |
| PanNuke | 9 | 623 | 1,888 | Tissue classification |
| WSSS4LUAD | 2 | 1,009 | 3,028 | Tissue classification |

Table 16: Comparison of Expected Calibration Error (ECE) on the ImageNet dataset. Lower values indicate better calibration. Note that CLIP achieves the lowest ECE without post-hoc calibration methods like Temperature Scaling (TS) or Vector Scaling (VS).

| Method | DenseNet 161 | DenseNet 161+TS | DenseNet 161+VS | ResNet 152 | ResNet 152+TS | ResNet 152+VS | **CLIP** |
|---|---|---|---|---|---|---|---|
| **ECE** | 6.28% | 1.99% | 2.24% | 5.48% | 1.86% | 2.23% | **1.51%** |

In the table, "TS" refers to Temperature Scaling, and "VS" refers to Vector Scaling. These are calibration strategies applied to DenseNet and ResNet to reduce Expected Calibration Error (ECE), whereas CLIP adopts no such strategies. Experimental data were obtained on the ImageNet dataset.

The table demonstrates that CLIP's ECE is significantly lower than that of DenseNet and ResNet. Even after applying scaling strategies to reduce ECE for the baseline models, CLIP's ECE remains lower, confirming its superior calibration performance.

## A.8 Statistical significance for results

To evaluate the stability and robustness of our proposed method, Dota, we conducted experiments across all datasets using five different random seeds. The aggregated results are presented in Tab. 17, Tab. 18, and Tab. 19, which detail the mean accuracy, standard deviation (stddev), and 95% confidence intervals. As shown, Dota exhibits highly consistent performance. This analysis confirms the robustness and reliability of our findings.

Table 17: Statistical analysis for cross-domain generalization results.

| Dataset | Aircraft | Caltech101 | Cars | DTD | EuroSAT | Flowers | Food101 | Pets | Sun397 | UCF101 | Average |
|---|---|---|---|---|---|---|---|---|---|---|---|
| Dota (seed=1) | 26.25 | 94.16 | 69.56 | 47.64 | 62.78 | 75.23 | 87.08 | 92.01 | 69.87 | 72.54 | 69.71 |
| Dota (mean) | 26.18 | 94.59 | 69.57 | 47.88 | 62.71 | 75.32 | 87.06 | 91.91 | 69.82 | 72.69 | 69.77 |
| stddev | 0.26 | 0.29 | 0.17 | 0.36 | 0.28 | 0.21 | 0.06 | 0.28 | 0.07 | 0.21 | 0.22 |
| confidence interval | [25.86, 26.51] | [94.22, 94.96] | [69.35, 69.78] | [47.44, 48.33] | [62.37, 63.06] | [75.06, 75.58] | [86.99, 87.13] | [91.56, 92.26] | [69.73, 69.91] | [72.43, 72.94] | [69.50, 70.05] |

## A.9 Implementation details

All the models in our experiments are built upon the pre-trained CLIP model [39] that consists of an image encoder and a text encoder. Test-time adaptation is set for single-image scenarios, using a batch size of 1. For natural distribution shifts scenario, we tune our hyperparameters using the validation set. For the cross-domain generalization scenario, we perform hyperparameter search using the corresponding validation sets. We adjust $\sigma^2$ within [0.001, 0.002, 0.004], then search for the best

Table 18: Statistical analysis for Natural Distribution Shift (NDS) results.

| Dataset | ImageNet | ImageNet-A | ImageNet-R | ImageNet-S | Average |
|---|---|---|---|---|---|
| Dota (seed=1) | 70.69 | 61.50 | 81.21 | 51.84 | 66.31 |
| Dota (mean) | 70.66 | 61.32 | 81.13 | 51.85 | 66.24 |
| stddev | 0.07 | 0.20 | 0.11 | 0.11 | 0.12 |
| confidence interval | [70.57, 70.75] | [61.06, 61.57] | [81.00, 81.26] | [51.71, 51.99] | [66.08, 66.39] |

Table 19: Statistical analysis for medical scenario results.

| Dataset | Kather | PanNuke | WSSS4LUAD | Average |
|---|---|---|---|---|
| Dota (seed=1) | 61.92 | 74.68 | 75.07 | 70.56 |
| Dota (mean) | 61.94 | 74.65 | 75.13 | 70.57 |
| stddev | 0.14 | 0.14 | 0.15 | 0.14 |
| confidence interval | [61.76, 62.11] | [74.49, 74.82] | [74.94, 75.32] | [70.40, 70.75] |

$\eta$ across [0.2, 0.3, 0.4, 0.5] and $\rho$ across [0.005, 0.01, 0.02, 0.03], with the shrinkage parameter $\epsilon$ set to 0.0001. We use top-1 accuracy (%) as our evaluation metric. All experiments are conducted using a single NVIDIA RTX 4090 GPU and a 12-core Intel Xeon Platinum 8352V CPU.

