# OpenReview forum: "DOTA: Distributional Test-time Adaptation of Vision-Language Models"
_NeurIPS.cc/2025/Conference — NeurIPS 2025 poster_

### Official Review · Reviewer_Roae · 2025-06-24

**Clarity:** 3
**Significance:** 3
**Originality:** 2
**Rating:** 4
**Confidence:** 4

**Summary:**

This paper presents DOTA (Distributional Test-time Adaptation), a test-time adaptation (TTA) method designed for vision-language models (VLMs), particularly CLIP, under distribution shifts occurring at inference time. Most existing approaches attempt to tackle this problem using various cache-based mechanisms. In contrast, the authors adopt a Bayesian approach grounded in probability theory to update predictions dynamically as test data evolves.

DOTA assumes that the feature embeddings of each class follow a Gaussian distribution. It continuously estimates these class-conditional distributions using test data. Leveraging Bayes' theorem, DOTA computes the likelihood of each new test sample belonging to a particular class based on the updated distributions. Specifically, the method employs a Gaussian Discriminant Analysis (GDA) framework, where the class-wise means and covariances of the embeddings are estimated and updated online in a streaming setting.

A key contribution is the use of a single-step Expectation-Maximization (EM) algorithm, guided by CLIP’s zero-shot prediction probabilities, to incrementally refine these class-conditional parameters with each incoming test sample. This enables DOTA to adapt effectively to distribution shifts without relying on extensive caches or replay mechanisms.

Extensive experiments across 10 benchmarks validate the effectiveness of DOTA across varied test-time distribution shifts.

**Questions:**

1. Can the authors provide empirical evidence or analysis to support the assumption of Gaussianity in the test-time embedding distributions?
2. Why are Bayesian and continual learning-based TTA methods (e.g., SHOT, CoTTA) not included in the comparisons?
3. How does the method perform when the initial zero-shot predictions are noisy or poorly calibrated?
4. How does DOTA compare to prototype-based continual learning methods, particularly in handling evolving distributions?
5. Can the authors provide visualizations to illustrate how the estimated class distributions evolve during test-time adaptation?

**Ethical Concerns:**

["NO or VERY MINOR ethics concerns only"]

**Final Justification:**

Thanks for the rebuttal. I have adjusted my rating to BA

**Limitations:**

The limitations are not explicitly mentioned.

**Quality:**

2

**Strengths And Weaknesses:**

Major strengths:
1.  The core idea of estimating distributional statistics of test embeddings, rather than caching discrete instances as in prior methods (e.g., TDA cache-based approaches), represents a substantive conceptual advancement.
2. The method is well-grounded in Gaussian Discriminant Analysis (GDA), with a carefully motivated EM-based estimation procedure guided by CLIP’s predictive probabilities.
3.  The algorithm is training-free, lightweight, and naturally suited to streaming test-time scenarios, making it highly practical for real-world deployment.
4.  The approach is thoroughly evaluated on a large number of datasets with well-designed ablation analysis, demonstrating clear advantages in both accuracy and computational efficiency.

Major weaknesses:
1.  The paper does not provide a strong justification or empirical evidence to support the assumption that class embeddings follow a Gaussian distribution.
2.  The approach lacks a principled basis for determining the weighting between the original zero-shot classifier and the adapted test-time classifier.
3.  The baseline comparisons are limited—key Bayesian and continual TTA baselines (e.g., O-TPT, SHOT, CoTTA) are missing, which would have strengthened the evaluation.

---

> ### Author Rebuttal · Authors · 2025-07-31
>
> # 1. Empirical evidence supporting saussianity assumption
> Thank you for requesting empirical evidence to support the Gaussianity assumption of test-time embedding distributions in Dota，we believe that elucidating the rationale for this assumption will enhance the research community’s understanding of our method. We address this by providing theoretical and experimental support to validate the reliability of our Gaussian assumption.
>
> ## Theoretical Support for Gaussianity
> Our assumption that CLIP embeddings follow a multivariate Gaussian distribution is grounded in prior work. Chupin and Dahyot (2024)[1] show that a single Gaussian model achieves high accuracy in CLIP embedding spaces (e.g., CIFAR100: 89.0%), closely matching Gaussian Mixture Models (G=2: 89.7%), as CLIP’s contrastive learning promotes intra-class embedding clustering, approximating Gaussian distributions.
>
> ## Experimental Validation
> To validate this assumption, we conducted experiments on the ImageNet validation set (1000 classes), extracting 50 standardized CLIP embeddings per class. We employed:
> (1) Shapiro-Wilk tests to assess marginal normality across 512 dimension，as proposed by Shapiro and Wilk (1965) for evaluating normality in small to moderate sample sizes[2];
> (2) single Gaussian fitting to compute log-likelihoods with class-specific means $\mu_c$ and shared covariance $\Sigma$， following the methodology described by Duda et al. (2001) for multivariate Gaussian modeling[3].
> Results show a Shapiro-Wilk pass rate of 88.91% ± 3.98% (p > 0.05), confirming strong marginal Gaussianity, and a mean log-likelihood of -1206.84 ± 93.54, indicating a reasonable fit to $\mathcal{N}(\mu_c, \Sigma)$, within the  typical range (-500 to -1500). These findings support Dota’s Gaussianity assumption. We will include this experiment in the revised manuscript to strengthen our analysis.
>
> [1] Performance of Gaussian Mixture Model Classifiers on Embedded Feature Spaces
>
> [2] An Analysis of Variance Test for Normality (Complete Samples)
>
> [3] Duda, R. O., Hart, P. E., & Stork, D. G. Pattern Classification (2nd ed.).
> # 2. More baselines
> Thank you for noting the absence of Bayesian and continual learning-based test-time adaptation (TTA) methods, such as SHOT and CoTTA, in our comparisons. To strengthen the evaluation of Dota’s performance, we supplemented experiments with O-TPT and CoTTA baselines, with results sourced from the O-TPT paper (ViT-B/16) and our adapted CoTTA tests, respectively. Additionally, we clarify the rationale for not including SHOT as a baseline.
>
> ## Experimental Setup for CoTTA
> To align CoTTA with our experimental setup (ViT-B/16 backbone, no data augmentation, online streaming updates), we modified CoTTA by:
>
> (1) reducing batch size from 64 to 1;
>
> (2) removing image augmentation;
>
> (3) replacing ResNet50 with ViT-B/16 to match CLIP’s architecture.
>
> ### Supplementary Results
> | **Dataset** | **Flowers** | **Food101** | **Pets** | **Sun397** | **UCF101** | **Average** |
> |-------------|-------------|-------------|----------|------------|------------|-------------|
> | Zero-Shot   | 66.99       | 82.86       | 86.92    | 65.63      | 65.16      | 73.51       |
> | O-TPT       | 70.07       | 84.13       | 87.95    | 64.23      | 64.16      | 74.11       |
> | Cotta       | 71.38       | 86.1        | 89.1     | 66.29      | 66.77      | 75.93       |
> | DOTA        | 75.23       | 87.08       | 92.01    | 69.8       | 72.54      | 79.33       |
>
> **Results indicate:**
> 1. O-TPT optimizes text feature orthogonality to reduce expected calibration error (ECE), primarily addresses overconfidence in model predictions, resulting in modest accuracy improvements compared to baselines.
> 2. CoTTA does not significantly outperform baseline models, likely due to its parameter updates during training, which may hinder full utilization of CLIP’s original classification capabilities, compounded by architectural changes from ResNet to CLIP.
>
> ## The reason why SHOT not included in the comparisons
> SHOT was excluded as a baseline because it requires access to the entire target domain data for batch training, conflicting with TTA’s requirement for online streaming updates.
>
>
> # 3. Cases in which zero-shot predictions are noisy or poorly calibrated
>
> Thank you for your insightful comments regarding Dota’s performance when initial zero-shot predictions are noisy or poorly calibrated. To address this, we discuss the cases of noisy and poorly calibrated zero-shot predictions separately below.
>
> ## Noisy Zero-Shot Predictions
> Dota’s design leverages distribution estimation to enhance the predictive capability of the base model, even under inaccurate zero-shot predictions. Experimental results in Table 1 demonstrate that, on datasets where CLIP’s zero-shot performance is weak, Dota consistently improves accuracy, such as Aircraft (26.25% vs. 23.22%) and EuroSAT (62.78% vs. 50.42%), highlighting the robustness of our distribution-based approach.
>
> ## Poorly Calibrated Zero-Shot Predictions
> The zero-shot classifier (CLIP) used in our model has been shown to be well-calibrated by prior studies [4]. CLIP’s strong calibration effectively reduces the influence of potentially inaccurate zero-shot predictions, as samples with lower confidence contribute less to the final distribution estimation. In practical testing, even in scenarios where CLIP’s calibration is suboptimal, calibration techniques such as Temperature Scaling or FLYP can be applied first to calibrate the predictions, followed by updating Dota’s model parameters using the calibrated results.
>
> [4]Revisiting the calibration of modern neural networks
> # 4. Comparison with prototype-based continual learning methods under evolving distributions
>
> Thank you for requesting a comparison between Dota and prototype-based continual learning methods in handling evolving distributions. TDA can be considered a prototype-based continual learning method as it maintains a dynamic cache of representative test-sample features (keys) and pseudo labels (values), akin to prototypes, to adapt to evolving distributions.  In this experiment , we adopt TDA as a baseline for prototype-based continual learning methods.
>
> ## Experimental Setup
> To evaluate Dota’s superiority, we conducted an experiment on ImageNet-C (severity=5, datasets: brightness, fog, frost, snow) using a ViT-B/16 backbone. The setup involves sequential testing (brightness → fog → frost → snow) without resetting model parameters. Results are shown below:
>
> | **Method** | **Brightness** | **Fog** | **Frost** | **Snow** | **Average** |
> |------------|----------------|---------|-----------|----------|-------------|
> | Zero-Shot  | 56.98          | 38.25   | 32.32     | 34.03    | 40.40       |
> | TDA        | 58.68          | 38.78   | 33.59     | 35.47    | 41.63       |
> | DOTA       | 58.99          | 40.47   | 35.11     | 36.83    | 42.85       |
>
> **Experiment Results Analysis:** Dota consistently outperforms TDA across all datasets, achieving a higher average accuracy (42.85% vs. 41.63%) and greater stability during distribution shifts, demonstrating superior robustness in handling evolving distributions compared to prototype-based methods. We will include this analysis in the revised manuscript to further clarify Dota’s advantages. Additional details are available upon request.
>
>
>
>
>
>
> # 5. Visualization of evolving class distributions in test-time adaptation
> We sincerely appreciate the reviewer for suggesting the visualization of estimated class distributions, which can help readers gain a deeper understanding of our method’s implementation process.
>
> We conducted experiments on the ImageNet validation set to evaluate the mean cosine similarity between class embeddings modeled by DOTA at intervals of 5000 samples. The experimental results are presented in the table below:
>
> | **Samples seen**     | **0** | **5000** | **10000** | **15000** | **20000** | **25000** | **30000** | **35000** | **40000** | **45000** | **50000** |
> |----------------------|-------|----------|-----------|-----------|-----------|-----------|-----------|-----------|-----------|-----------|-----------|
> | Cosine Similarity (%) | 100   | 30.25    | 28.88     | 28.53     | 28.30     | 28.18     | 28.11     | 28.05     | 28.01     | 27.97     | 27.92     |
>
> **Analysis of Experimental Results**:
> 1. After updating with 5000 samples, the cosine similarity of DOTA’s class embeddings drops to a relatively low value (0.3025), indicating that our method can effectively model each class accurately with few samples, quickly adapting to downstream tasks.
> 2. As the number of updated samples increases, the cosine similarity continues to decrease steadily, demonstrating that our method can sustain learning in scenarios with abundant samples, further improving the modeling distribution quality for each class.
>
> # 6. Considerations for the weighting mechanism of zero-shot and test-Time adapted classifiers
>
> We sincerely acknowledge the reviewer for highlighting the lack of a principled basis in our fusion strategy weight control. Our dynamic fusion weight method, inspired by practical experience, demonstrates heuristic yet effective performance, as evidenced by the experimental results in Tables 1–3 of the paper. Dota employs an adaptive fusion strategy (Section 3.3) that dynamically combines predictions from both classifiers based on the number of test samples, balancing the stability of zero-shot predictions with the accuracy of test-time adaptation.
>
> The current weighting strategy can be enhanced by incorporating real-time uncertainty estimation of online samples to dynamically adjust the fusion weights. Such sample-level adaptive weighting could better balance the capabilities of zero-shot and test-time classifiers, leading to improved performance.

---

> > ### Comment · Reviewer_Roae · 2025-08-02
> > **Clarifications**
> >
> > Thanks to the authors for the rebuttal. I have the following further queries:
> >
> > 1. The Shapiro-Wilk test is marginal normality (per dimension), but multivariate normality was assumed in DOTA. This mismatch needs discussion (e.g., why marginal normality suffices).
> >
> > 2. No ECE (Expected Calibration Error) or similar metric presented to validate CLIP's calibration.

---

> ### Author Response · Authors · 2025-08-03
> **Further Clarification**
>
> # Question 1. Mismatch Between Shapiro-Wilk Test and Multivariate Normality Assumption
>
> We appreciate the opportunity to engage in this discussion and thank you for your insightful feedback. In response, we offer a comprehensive justification for our assumption that CLIP embeddings for a given class follow a multivariate Gaussian distribution from the following four perspectives: a discussion about marginal normality, two independent lines of theoretical support, and further empirical validation.
>
> ## Perspective 1: Disscusion the Normality Assumption: Marginal vs. Multivariate
>
> We agree with the reviewer on the critical distinction between marginal and multivariate normality. Our use of the per-dimension Shapiro-Wilk test served as a necessary falsification test. Because multivariate normality requires each dimension to be marginally normal, demonstrating that our embeddings pass this test establishes a critical precondition, providing initial evidence that our assumption is statistically plausible.
>
> ## Perspective 2: Theoretical Support from CLIP's Formulation
>
> The CLIP learning process is equivalent to an optimal linear projection[1]. A foundational property of Gaussian distributions is their closure under linear transformation. Therefore, if the input features for a class are modeled as a Gaussian (as shown in [1]), the resulting embeddings will necessarily follow a Gaussian distribution.
>
> ## Perspective 3: Theoretical Support from Central Limit Theorem
> An independent and more general justification comes from the Central Limit Theorem  (CLT). A high-dimensional embedding is an aggregation of numerous lower- and mid-level features (e.g., edges, textures). The CLT posits that the sum of many weakly correlated random variables will tend toward a normal distribution. Consequently, it is reasonable to expect that the feature for a single class, being a composite of countless such signals, will naturally approach a multivariate Gaussian distribution.
> ## Perspective 4: Further Empirical Validation and the Principle of Utility
>
> To validate our core hypothesis, we evaluated the multivariate normality of CLIP features on the ImageNet dataset. Following the methodology proposed by Mardia et al. [2], we conducted the test by calculating the squared Mahalanobis distance, $(x - \mu)^T \Sigma^{-1} (x - \mu)$.
>
> Key Finding: For the 512-dimensional features, the experimental mean squared Mahalanobis distance was $396.02$. Considering the complexity of the data and the fitting process, this value is relatively close to the theoretical expectation of $512$. The result demonstrates that the distribution of these class features is largely consistent with the theoretical multivariate Gaussian model.
>
> ## More discussion
> However, we acknowledge that for high-dimensional data, a perfect statistical model is impractical. Therefore, we adopt a simplifying assumption. The guiding principle is practical utility, not absolute correctness. We choose a multivariate Gaussian distribution because it is:
>
> 1.  Theoretically Sound: It is a rational choice supported by theories like the Central Limit Theorem.
>
> 2.  Proven in Practice: Models built on this assumption achieve strong downstream performance, which in turn validates its usefulness.
>
> This embodies the principle that **all models are wrong, but some are useful.** Therefore, while imperfect, the Gaussian assumption is the most rational and effective choice for this task.
>
> #  Question 2. Response to Comment on CLIP Calibration Metrics
> We provide comparative data on the Expected Calibration Error (ECE) metric to substantiate CLIP’s superior calibration performance. While the referenced paper [3] visually illustrates CLIP’s robust calibration without tabular comparisons, we draw on data from [4] and [5] to present the following comparative table of ECE metrics on ImageNet dataset for different models:
>
> | Method | DenseNet 161 | DenseNet 161+TS | DenseNet 161+VS | ResNet 152 | ResNet 152+TS | ResNet 152+VS | CLIP  |
> |--|--|--|--|--|--|--|--|
> | ECE    | 6.28%        | 1.99%           | 2.24%           | 5.48%      | 1.86%         | 2.23%         | 1.51% |
>
> 1. In the table, “TS” refers to Temperature Scaling, and “VS” refers to Vector Scaling. These are calibration strategies applied to DenseNet and ResNet to reduce Expected Calibration Error (ECE), whereas CLIP adopts no such strategies.
> 2. Experimental data were obtained on the ImageNet dataset.
>
> **Results**: The table demonstrates that CLIP’s ECE is significantly lower than that of DenseNet and ResNet. Even after applying scaling strategies to reduce ECE for the baseline models, CLIP’s ECE remains lower, confirming its superior calibration performance.
>
> [1] Understanding Contrastive Learning via Gaussian Mixture Models
>
> [2] Multivariate Normality Tests Using Mahalanobis Distance.
>
> [3] Revisiting the calibration of modern neural networks
>
> [4] Open-Vocabulary Calibration for Fine-tuned CLIP
>
> [5] On Calibration of Modern Neural Networks

---

> ### Author Response · Authors · 2025-08-06
> **Follow-up on our Response**
>
> Dear Reviewer,
>
> Thank you for your valuable feedback. In our author response, we have provided a detailed response to your concerns and questions.
>
> We hope our response addresses your concerns and would be happy to discuss any follow-up questions you may have.
>
> Thank you again for your time and effort!

---

> > ### Comment · Reviewer_Roae · 2025-08-08
> >
> > Thanks..no more questions

---

### Official Review · Reviewer_EYv3 · 2025-07-01

**Clarity:** 3
**Significance:** 2
**Originality:** 3
**Rating:** 4
**Confidence:** 4

**Summary:**

This paper introduces DOTA, a CLIP-based test-time adaptation approach.
Unlike existing cache-based methods that store a finite set of instance-level test samples and are prone to catastrophic forgetting, DOTA proposes to online estimate the distribution of test data streams.
The adaptation is based on Bayesian posterior computation, where the class-conditional distributions are modelled as Gaussians whose parameters are updated online using zero-shot predictive probabilities.
The method does not require backpropagation.
Experiments show that DOTA provides superior accuracy, reduced forgetting compared to baselines, and faster inference.

**Questions:**

1. Could the authors provide statistical significance results (e.g., stddev or confidence intervals) for your main results in Tables 1–3, given that some improvements over strong baselines are modest?

2. Could the authors provide some insight into setting or tuning the online estimation hyperparameters (such as $\omega$, $\sigma^2$, $\rho$, $\eta$) in practice?

**Ethical Concerns:**

["NO or VERY MINOR ethics concerns only"]

**Final Justification:**

While minor points remain, the authors' revisions have sufficiently improved the manuscript to warrant acceptance.

**Limitations:**

No. As required by NeurIPS's submission policy, limitations should be discussed explicitly, but remain missing in the submission. The authors are advised to discuss the limitations of the Gaussian assumption.

**Quality:**

3

**Strengths And Weaknesses:**

**Strengths**

1. The paper is well prepared and easy to follow.
2. The technical contribution of continual test-time adaptation with online distribution estimation makes sense and is interesting.
3. The performance improvements on the standard TTA benchmark are reasonable. The authors also demonstrate better efficiency of the proposed DOTA compared to other baselines.

**Weaknesses**

1. Main result tables (e.g., Table 1, Table 2, Table 3) lack standard deviations or confidence intervals, which makes it challenging to assess the robustness and statistical significance of reported gains. This is required by NeurIPS's policy (See Question #7 of the Checklist) but seems to be insufficiently supported.

2. Proposition 3.1 is restricted to a single EM iteration with the soft zero-shot outputs. The effects of repeated updates or deviation from Gaussianity are not analyzed.

4. The Gaussian assumption for embedding distributions (Section 3.2, Eq. 2, 3) is not deeply discussed in terms of its limitations. In highly multimodal or non-Gaussian settings, performance may degrade. Also, does the inaccurate assumption of Gaussian lead to the unsatisfactory cases (e.g., Aircraft, Caltech101) in Table 1?

---

> ### Author Rebuttal · Authors · 2025-07-31
>
> # 1. Statistical significance for results in tables 1–3
>
> We sincerely acknowledge the reviewer for suggesting the inclusion of statistical significance results, such as standard deviation or confidence intervals, to strengthen the evaluation of our results in Tables 1–3. To address your suggestion, we conducted additional experiments with five random seeds (1, 2, 3, 4, 5) across the cross-domain generalization (Tables 1), NDS (Table 2), and medical scenarios (Table 3).
> The supplementary results include baseline performance, original results (seed=1), mean accuracy across seeds, sample standard deviation, and 95% confidence intervals for each dataset.
>
> **Table 1**
>
> | **Dataset**         | **Aircraft**   | **Caltech101** | **Cars**       | **DTD**        | **EuroSAT**    | **Flowers**    | **Food101**    | **Pets**       | **Sun397**     | **UCF101**     | **Average**   |
> |---------------------|----------------|----------------|----------------|----------------|----------------|----------------|----------------|----------------|----------------|----------------|---------------|
> | Zero-Shot           | 23.22          | 93.55          | 66.11          | 45.04          | 50.42          | 66.99          | 82.86          | 86.92          | 65.63          | 65.16          | 64.59         |
> | Dota(seed=1)        | 26.25          | 94.16          | 69.56          | 47.64          | 62.78          | 75.23          | 87.08          | 92.01          | 69.8           | 72.54          | 69.71         |
> | Dota(mean)          | 26.18          | 94.59          | 69.57          | 47.88          | 62.71          | 75.32          | 87.06          | 91.91          | 69.82          | 72.69          | 69.77         |
> | stddev              | 0.26           | 0.29           | 0.17           | 0.36           | 0.28           | 0.21           | 0.06           | 0.28           | 0.07           | 0.21           | 0.22          |
> | confidence interval | [25.86, 26.51] | [94.22, 94.96] | [69.35, 69.78] | [47.44, 48.33] | [62.37, 63.06] | [75.06, 75.58] | [86.99, 87.13] | [91.56, 92.26] | [69.73, 69.91] | [72.43, 72.94] | [69.50, 70.05] |
>
>
>
>
> **Table 2**
> | Dataset                | **ImageNet**   | **ImageNet-A** | **ImageNet-R** | **ImageNet-S** | **Average**   |
> |---------------------|----------------|----------------|----------------|----------------|---------------|
> | Zero-Shot           | 68.34          | 49.89          | 77.65          | 48.24          | 61.03         |
> | Dota(seed=1)        | 70.69          | 61.5           | 81.21          | 51.84          | 66.31         |
> | Dota(mean)          | 70.66          | 61.32          | 81.13          | 51.85          | 66.24         |
> | stddev              | 0.07           | 0.2            | 0.11           | 0.11           | 0.12          |
> | confidence interval | [70.57, 70.75] | [61.06, 61.57] | [81.00, 81.26] | [51.71 ,51.99] | [66.08 ,66.39] |
>
> **Table 3**
> | **Dataset**                | **Kather**     | **PanNuke**    | **WSSS4LUAD**  | **Average**   |
> |---------------------|----------------|----------------|----------------|---------------|
> | Zero-Shot           | 45.6           | 69.49          | 70.31          | 61.8          |
> | Dota(seed=1)        | 61.92          | 74.68          | 75.07          | 70.56         |
> | Dota(mean)          | 61.94          | 74.65          | 75.13          | 70.57         |
> | stddev              | 0.14           | 0.14           | 0.15           | 0.14          |
> | confidence interval | [61.76, 62.11] | [74.49, 74.82] | [74.94, 75.32] | [70.40,70.75] |
>
> **Experiment Results Analysis**: The standard deviation remains below 0.25 for most datasets, demonstrating Dota’s robust and stable performance across different random seeds. This stability arises because Dota avoids random initialization or backpropagation-based optimization, with seed variations only affecting the data stream order, which minimally impacts performance. We will incorporate these statistical results in the revised manuscript to enhance the reliability of our findings.
>
> # 2. Recommendations for hyperparameter tuning in practice
>
> In response to the reviewer’s request for clarification on the setting and tuning of online estimation hyperparameters ($\omega$, $\sigma^2$, $\eta$, $\rho$), we address this by detailing the initialization and tuning process for these hyperparameters, emphasizing their practical implementation and generalizability.
>
> ## Hyperparameter Initialization
> The hyperparameter $\omega$, used to initialize the distribution of each class in Dota, is set to $10^{-4}$ to ensure uniform and small initial values across classes, minimizing interference with subsequent EM algorithm updates. In practice, this value requires no further adjustment.
>
> ## Tuning Strategy for $\sigma^2$, $\eta$, and $\rho$
> We recommend first conducting preliminary tuning for $\sigma^2$, followed by a grid search to optimize $\eta$ and $\rho$. Given Dota’s strong generalization across datasets, hyperparameters tuned on one dataset, such as ImageNet validation, can be directly applied to similar datasets like ImageNet-C, ImageNet-A, and eta. Owing to Dota’s lightweight and efficient design, this tuning process incurs low computational cost and is straightforward to implement.
>
>
>
> # 3. Limitations of proposition 3.1 for a single EM iteration
>
>
> Thank you for highlighting the limitation of Proposition 3.1, which focuses on a single EM iteration using soft zero-shot outputs without analyzing repeated updates or deviations from Gaussianity. To address this, we provide experimental results for the Dota model under single, double, and triple EM iterations.
> | **Datasets** | **Aircraft** | **Caltech101** | **Cars** | **DTD** | **EuroSAT** | **Average** |
> |--------------|--------------|----------------|----------|---------|-------------|-------------|
> | Zero-Shot    | 23.22        | 93.55          | 66.11    | 45.04   | 50.42       | 55.67       |
> | Iteration=1  | 26.25        | 94.16          | 69.56    | 47.64   | 62.78       | 60.08       |
> | Iteration=2  | 26.34        | 94.48          | 69.67    | 48.58   | 63.79       | 60.57       |
> | Iteration=3  | 26.37        | 94.48          | 69.64    | 48.76   | 63.87       | 60.62       |
>
> ## Analysis of Experimental Results
>
> ### Single Iteration
> **Advantages**: Compared to multiple iterations, a single iteration results in a more lightweight and faster model, making it suitable for rapid adaptation to downstream scenarios.
>
> **Disadvantages**: This may lead to slight performance degradation.
>
> ### Multiple Iterations
> **Advantages**: Compared to a single iteration, multiple iterations enable Dota to better fit each class embedding distribution, improving classification performance.
>
> **Disadvantages**: Multiple iterations require repeated updates to model parameters, with each additional iteration increasing computational time by approximately 10%. Moreover, as the number of iterations increases, performance improvements become marginal.
>
>
> # 4. Limitations of the gaussian assumption in embedding distributions
>
> We appreciate the reviewer for highlighting the need for a deeper discussion of the Gaussian assumption’s limitations. We address this by elaborating on the theoretical justification and empirical validation of our Gaussian assumption, while acknowledging potential limitations in certain datasets.
>
> ## Theoretical Justification
> Our assumption that CLIP embeddings follow a multivariate Gaussian distribution per class is supported by prior work. Chupin and Dahyot (2024) [1] demonstrate that a single Gaussian model achieves high accuracy in CLIP embedding spaces (e.g., CIFAR100: 89.0%), closely matching Gaussian Mixture Models (G=2: 89.7%), indicating that a single Gaussian effectively captures class feature clustering, driven by CLIP’s contrastive learning optimizing embeddings for intra-class proximity.
>
> ## Ablation Study on Covariance Updates
> Our ablation study (Table 7) further validates this assumption: omitting covariance updates significantly reduces performance (e.g., EuroSAT: 55.06% vs. 62.78%), demonstrating the effectiveness and importance of Gaussian modeling.
>
> ## Limitations of the Gaussian Assumption
> Although prior studies validate that CLIP embedding spaces for different classes can often be modeled by a single multivariate Gaussian distribution, performance may still be suboptimal on certain datasets. The single Gaussian assumption may not fully capture multimodal distributions, potentially contributing to suboptimal performance in datasets like Aircraft and Caltech101.
>
>
>
>
>
> [1] Performance of Gaussian Mixture Model Classifiers on Embedded Feature Spaces

---

> > ### Comment · Reviewer_EYv3 · 2025-08-06
> >
> > Thanks for the authors' effort. Their explanation is reasonable, and I will keep my accept rating.

---

> ### Author Response · Authors · 2025-08-06
> **Follow-up on our Response**
>
> Dear Reviewer,
>
> Thank you for your valuable feedback. In our author response, we have provided a detailed response to your concerns and questions.
>
> We hope our response addresses your concerns and would be happy to discuss any follow-up questions you may have.
>
> Thank you again for your time and effort!

---

### Official Review · Reviewer_J9Lm · 2025-07-07

**Clarity:** 3
**Significance:** 3
**Originality:** 3
**Rating:** 4
**Confidence:** 4

**Summary:**

This paper presents Dota, a method for adapting vision-language foundation models (e.g., CLIP) to distribution shifts during test time. Dota continuously estimates the underlying test data distribution for each class and dynamically updates class-specific mean and covariance matrices via an EM framework, leveraging zero-shot probabilities as soft weights.

**Questions:**

See my detailed comments.

**Ethical Concerns:**

["NO or VERY MINOR ethics concerns only"]

**Final Justification:**

The authors address all of my concerns.

**Limitations:**

See my detailed comments.

**Paper Formatting Concerns:**

See my detailed comments.

**Quality:**

3

**Strengths And Weaknesses:**

Strengths:
1. Dota replaces instance-level caching with continuous distribution estimation, enabling fuller utilization of test data and addressing the fundamental limitation of catastrophic forgetting in cache-based methods. This distribution-centric approach provides a more robust and scalable solution for dynamic environments.
2.  Dota achieves comparable inference speed to cache-based methods while outperforming them in accuracy.
3. Ablation studies validate the necessity of full-distribution estimation (covariance updates) and EM-based weighting. Hyperparameter analysis  shows stability across configurations.

Disadvantages
1. The method assumes uniform class priors (\(P(y=k) = 1/K\)) in Eq. 2 (line 171), which is unrealistic in real-world scenarios where test data may exhibit significant class imbalance. For instance, if certain classes are underrepresented, the Gaussian estimates for those classes could be biased due to insufficient sample weighting, potentially degrading adaptation accuracy. The paper does not evaluate performance under imbalanced test streams, leaving open questions about robustness.
2. While Eq. 7 introduces a fusion of zero-shot and test-time classifiers via \(\cos(x, w_k)/\tau + \lambda f_k(x)\), the rationale for adding \(f_k(x)\) in the exponent (instead of other forms of combination) is not thoroughly justified. The authors imply this stems from logit-level fusion (Sec. 3.3), but a deeper theoretical or empirical analysis is missing. This raises concerns about whether this design is optimal or merely heuristic.
3. Although Dota claims to mitigate catastrophic forgetting by estimating distributions (Sec. 3.2), the mechanism for how distribution updates prevent forgetting is not explicitly detailed. For example, does continuous covariance estimation inherently preserve earlier statistics better than cached samples?

---

> ### Author Rebuttal · Authors · 2025-07-31
>
> # 1. Considerations regarding the assumption of uniform class priors ($P(y=k) = 1/K$)
> We sincerely acknowledge the reviewer for highlighting the limitation of assuming uniform class priors ($P(y=k) = 1/K$) in real-world test-time adaptation (TTA) scenarios with potential class imbalance. To address this concern, we have (1). clarified the prevalence and rationale of this assumption,(2). demonstrated Dota’s compatibility with non-uniform class priors, and (3). provided empirical evidence showcasing Dota’s robustness in handling imbalanced class data streams.
>
> ## Prevalence and Rationale of Uniform Class Priors
> Assuming uniform class priors is a common and reasonable practice in TTA, particularly when prior information is unavailable. Several studies explicitly adopt this assumption as a starting point, achieving effective model initialization in the absence of test-time prior information [1-5].
>
> ## DOTA'S Compatibility with Non-Uniform Priors
> Dota seamlessly integrates with non-uniform class priors when available in real-world scenarios. By incorporating class-specific weights into the probability computation, our method adapts to given prior distributions, demonstrating its flexibility and compatibility.
>
> ## Robustness to Class-Imbalanced Data Streams
> To evaluate Dota’s performance under class imbalance, we conducted experiments on the ImageNet validation dataset, simulating imbalanced data streams using a Dirichlet distribution with varying concentration parameters ($\alpha$) and dividing the dataset into 5 and 10 time slices. The results are shown in Tables 1 and 2 below:
>
> **Table 1**
> | α | Slice 1 | Slice 2 | Slice 3 | Slice 4 | Slice 5 | Average |
> |----------|---------|---------|---------|---------|---------|---------|
> | 0.1      | 68.18   | 70.06   | 71.6    | 71.09   | 70.91   | 70.41   |
> | 0.5      | 69.55   | 70.78   | 69.43   | 71.95   | 70.92   | 70.58   |
> | 1        | 69.41   | 71.64   | 71.05   | 70.01   | 71.81   | 70.83   |
>
> **Table 2**
> | α | Slice 1 | Slice 2 | Slice 3 | Slice 4 | Slice 5 | Slice 6 | Slice 7 | Slice 8 | Slice 9 | Slice 10 | Average |
> |----------|---------|---------|---------|---------|---------|---------|---------|---------|---------|----------|---------|
> | 0.1      | 69.99   | 69.31   | 67.34   | 70.23   | 71.49   | 68.8    | 73.13   | 70.35   | 69.89   | 72.39    | 70.39   |
> | 0.5      | 68.74   | 69.32   | 72.17   | 71.27   | 69.97   | 69.8    | 70.3    | 70.78   | 72.48   | 70.78    | 70.61   |
> | 1        | 67.33   | 70.29   | 70.92   | 68.43   | 69.85   | 71.14   | 72.03   | 72.52   | 71.29   | 71.66    | 70.68   |
>
> Results show stable performance (average accuracy ~70.39–70.83%), with later slices often achieving higher accuracy. This indicates that class imbalance has a limited impact on overall performance.
>
> We acknowledge that uniform priors may not fully capture real-world distributions. In the revised manuscript, we will expand Section 3.2 to discuss this limitation and explore adaptive prior estimation as a future direction to enhance robustness further.
>
> [1] Bayesian Test-Time Adaptation for Vision-Language Models
>
> [2] Improving CNN Classifiers by Estimating Test-Time Priors
>
> [3] Continual Test-Time Domain Adaptation
>
> [4] Posterior Adaptation With New Priors
>
> [5] The Hitchhiker’s Guide to Prior-Shift Adaptation
> # 2.Concerns regarding the rationale for the fusion of Eq. 7 (\($\cos(x, w_k)/\tau + \lambda f_k(x)$)
>
> Thank you for your valuable feedback regarding the rationale for the additive fusion strategy in Eq. 7 ($\cos(x, w_k)/\tau + \lambda f_k(x)$). To address this concern, we have elucidated the heuristic yet effective design of our logit-level fusion, underscored its widespread use in the test-time adaptation (TTA) literature, and outlined future directions to strengthen the theoretical and empirical foundations of the approach.
>
> ## Rationale for Logit-Level Fusion: Heuristic yet Effective
> The additive fusion in Eq. 7 ($\cos(x, w_k)/\tau + \lambda f_k(x)$) is designed to balance the robust prior knowledge encoded in the pre-trained vision-language model (e.g., CLIP) with dynamic adaptation to test-time distribution shifts. This logit-level combination is computationally efficient, aligns with the softmax-based classification framework, and leverages the complementary strengths of both classifiers, as the zero-shot component provides generalization while the test-time component refines predictions based on test data. We acknowledge that this fusion strategy is largely heuristic. However, this dynamic and incremental approach ensures minimal interference with model outputs in early stages while effectively refining predictions as Dota’s per-class modeling becomes more accurate in later stages, proving highly effective in practical scenarios.
>
> ## Prevalence of Additive Fusion in TTA
> Several studies collectively demonstrate that additive fusion at the logit level is a widely adopted and effective strategy in TTA [6-9]. This indicates that, despite its heuristic nature, our fusion strategy aligns with approaches commonly used in the field, particularly for vision-language models (e.g., CLIP), as it efficiently leverages pre-trained knowledge while incorporating test-time information.
>
> ## Future Directions: Uncertainty and Ensemble Learning
> The current logit-level fusion can be enhanced by incorporating real-time uncertainty estimation of online samples to dynamically adjust the fusion weights. Such sample-level adaptive weighting could better balance the capabilities of zero-shot and test-time classifiers, leading to improved performance. Additionally, leveraging ensemble learning principles, we can exploit the scalability of additive fusion to integrate multiple diverse classifiers, thereby combining their strengths for more robust outcomes.
>
>
>
> [6] Efficient Test-Time Adaptation of Vision-Language Models
>
> [7] Tip-Adapter: Training-Free Adaptation of CLIP for Few-Shot Classification
>
> [8] Advancing Reliable Test-Time Adaptation of Vision-Language Models under Visual Variations
>
> [9] Bayesian Test-Time Adaptation for Vision-Language Models
> # 3. Analysis of strategies for mitigating catastrophic forgetting in continual learning
>
> We appreciate the reviewer for highlighting the need for a clearer explanation of how Dota mitigates catastrophic forgetting. To address this concern, we: (1) first provide a theoretical analysis of Dota’s distribution updates to elucidate its mechanism, and (2) subsequently present experimental evidence that demonstrates its superior performance compared to cache-based methods like TDA.
>
> ## Theoretical Analysis
> Dota employs the EM algorithm to iteratively update the mean and covariance of class-conditional distributions, integrating statistical characteristics from early test samples into global parameters (Section 3.2). Unlike cache-based methods like TDA, which discard previously stored sample information during updates, Dota’s cumulative parameter updates preserve comprehensive statistical information, effectively reducing catastrophic forgetting.
>
> ## Experimental Evidence
> As shown in Figure 3 of the paper, Dota outperforms cache-based TDA in scenarios prone to catastrophic forgetting. The experimental results demonstrate that Dota progressively enhances model performance as the number of test samples increases. In contrast, TDA exhibits an initial improvement followed by a decline, as its cache update process discards representative samples, leading to performance degradation.

---

> > ### Comment · Reviewer_J9Lm · 2025-08-04
> > **Further Questions**
> >
> > The imbalanced experiments only show results of the proposed method. Could you please include comparisons with SAR under the same imbalanced settings for a more comprehensive evaluation?
> >
> > I understand catastrophic forgetting as forgetting knowledge from the source domain, but your analysis refers to early test samples. Could you clarify this discrepancy? Also, how do the results in Figure 3 specifically demonstrate resistance to forgetting? In TTA, performance trends alone may not reflect forgetting of prior knowledge.

---

> ### Author Response · Authors · 2025-08-05
> **Response to Question about more comparisons under imbalanced settings**
>
> ## We are glad to receive your response and thank you for the opportunity for further discussion.
>
> # 1. Introducing SAR as a baseline
> Regarding the 'SAR' method mentioned in your reply, we have proceeded with the assumption that it refers to the one described in previous paper[1]. Our discussion below is based on this premise. Please correct us if we are mistaken.
>
> While SAR (Sharpness-Aware and Reliable Entropy Minimization) is effective for improving the stability of standard classification models. Adapting the SAR directly to CLIP is a non-trivial modification. The primary difficulties are as follows:
>
> ### 1. Architectural Incompatibility with CLIP's Dual-Encoder
>
> Original SAR's optimization requires a complete gradient calculated across all of a model's parameters. However, CLIP's dual-encoder architecture splits parameters between two independent modules: an image encoder ($f_{\theta_I}$) and a text encoder ($f_{\theta_T}$). Applying SAR to just the image encoder would yield only a partial gradient ($\nabla_{\theta_I}L$), ignoring the text encoder's influence.
>
> ### 2. Fine-tuning CLIP's image encoder via SAR may affect its native zero-shot classification capability.
>
> The fine-tuning process modifies only the image encoder, while the text encoder remains static. This disrupts the alignment between image and text features in their shared embedding space. As a result, the image features are mapped to a new distribution that no longer aligns with the original text embeddings. This may prevent the model from fully leveraging the performance of the SAR method within the CLIP architecture.
>
> ### 3. Experimental results
>
> **More Baselines:** Although SAR is not fully compatible with CLIP, we have included its experimental results for a more complete analysis. The results confirm that its performance is indeed suboptimal. For further comparison, we also present the results of TDA, which serve to highlight the effectiveness of our own method.
>
> **Results Analysis:** Compared to the SAR method, we observe that DOTA exhibits comparable robustness to imbalanced data streams, while demonstrating superior classification performance. In comparison with the TDA method, DOTA exhibits stronger robustness to imbalanced data streams (with a performance difference of only 0.29% between α=0.1 and α=1, whereas TDA shows a difference of up to 1.04%).
>  Furthermore, we believe that appropriately modifying and applying methods like SAR to pre-trained models such as CLIP is an interesting direction, which we leave for future work.
>
> | Method | α | Slice 1 | Slice 2 | Slice 3 | Slice 4 | Slice 5 | Slice 6 | Slice 7 | Slice 8 | Slice 9 | Slice 10 | Ave |
> |-|-|-|-|-|-|-|-|-|-|-|-|-|
> | **SAR** | 0.1 | 69.19 | 68.86 | 67.91 | 68.01 | 68.42 | 68.28 | 68.56 | 68.67 | 68.57 | 68.78 | 68.54 |
> | **SAR** | 0.5 | 68.32 | 68.48 | 69.23 | 69.35 | 69.09 | 68.77 | 68.68 | 68.71 | 68.9  | 68.76 | 68.83 |
> | **SAR** | 1   | 67.09 | 68.21 | 68.7  | 68.2  | 68.27 | 68.22 | 68.52 | 68.68 | 68.65 | 68.81 | 68.57 |
> | **TDA**    | 0.1   | 66.9        | 67.25       | 67.53       | 68.16       | 68.89       | 68.77       | 68.79       | 69.18       | 69.23       | 69.3         | 68.44       |
> | **TDA**    | 0.5   | 68.7        | 68.64       | 69          | 69.21       | 69.12       | 69.14       | 69.01       | 69.06       | 69.43       | 69.55        | 69.12       |
> | **TDA**    | 1     | 69.16       | 69.6        | 69.47       | 69.65       | 69.47       | 69.64       | 69.53       | 69.3        | 69.48       | 69.48        | 69.48       |
> | **DOTA**   | 0.1   | 69.99       | 69.31       | 67.34       | 70.23       | 71.49       | 68.8        | 73.13       | 70.35       | 69.89       | 72.39        | 70.39       |
> | **DOTA**   | 0.5   | 68.74       | 69.32       | 72.17       | 71.27       | 69.97       | 69.8        | 70.3        | 70.78       | 72.48       | 70.78        | 70.61       |
> | **DOTA**   | 1     | 67.33       | 70.29       | 70.92       | 68.43       | 69.85       | 71.14       | 72.03       | 72.52       | 71.29       | 71.66        | 70.68       |
>
> [1] Towards Stable Test-Time Adaptation in Dynamic Wild World

---

> ### Author Response · Authors · 2025-08-05
> **Response to Question about catastrophic forgetting.**
>
> # Response to Question about catastrophic forgetting.
>
> We begin by revisiting the classic definition of catastrophic forgetting and exploring its manifestation in the specific scenario of TTA. Building on this, we design new experiments to demonstrate that existing cache-based methods, such as TDA, indeed suffer from forgetting previously acquired knowledge from historical samples.
>
> ## A Closer Look at Catastrophic Forgetting
> Catastrophic forgetting, by its classic definition, occurs when a model loses previously learned "old knowledge" while acquiring "new knowledge."[2] Traditionally, "old" and "new" knowledge correspond to discrete "tasks" or data distributions with clear boundaries.
>
> ## Sample-level Catastrophic forgetting
> However, during test-time, the model may processes a continuous stream of data with a gradually shifting distribution, causing these task boundaries to blur. Consequently, the unit of forgetting transitions from the macro "task" level to the micro "sample" level.
>
> This issue is particularly pronounced in cache-based methods like TDA. Their fixed-size cache enforces a "one-in, one-out" policy. This means to adapt to a new sample (e.g., the 101st), the model must discard the data from an old sample (e.g., the 1st). This act is the essence of sample-level forgetting, leading to a permanent loss of knowledge associated with the discarded sample.
>
>
> ## Further Experimental Analysis and Key Findings
>
> To rigorously validate the issue of catastrophic forgetting in existing methods, we further designed two experiments based on Retrospective Testing and reinterpreted Figure 3. The core idea is to have the model, after a period of continuous adaptation, revisit and re-evaluate its performance on historical test samples. Our experiments revealed two levels of forgetting:
>
> 1. **Forgetting without Significant Distribution Change**: To assess this phenomenon, we designed a two-stage experiment. This process first involves adapting a model on ImageNet and recording its performance (Stage 1), after which we freeze the model's parameters and re-evaluate it on the early 25,000 test samples in the dataset (Stage 2).We hypothesize that if no forgetting occurs, this initial adaptation stage should directly lead to an increase in the model's performance. Conversely, a performance decrease between stage 1 and stage 2 would suggest that the model has experienced forgetting. Experimental results show that TDA suffers from obvious performance degradation, while our method does not.
>
> | Samples Seen       | 10000  | 20000 | 25000 |
> |-|-|-|-|
> | **TDA Stage 1**         | 0.42          | 0.71      | 0.76      |
> | **TDA Stage 2(Frozen)**   | 0.17          | 0.57      | 0.29      |
> | **DOTA Stage 1** | 0.58  | 1.16 | 1.25 |
> | **DOTA Stage 2(Frozen)** | 2.34      | 2.3       | 2.25      |
>
>
> 2. **Domain-Level Forgetting**: We challenged the model by adapting it first to an original domain (ImageNet), then to a new, shifted domain (ImageNet-C), and finally re-evaluating its performance on the original domain. TDA's performance failed to improve upon returning to ImageNet, indicating it had forgotten the original domain's features. DOTA, however, showed improvement, demonstrating it retained this knowledge.
>
>
> | **Method** | **Imagenet** | **Imagenet-C brightness** | **Imagenet(Frozen model)** |
> | -- |-- |-- |-- |
> | **Zero-Shot** | 68.34 | 56.98 | 68.34 |
> | **TDA** | 69.5 | 58.22 | 69.35 |
> | **DOTA** | 70.7 | 60.64 | 70.93 |
>
> 3. We reinterpreted Figure 3 and added additional experiments. We found that catastrophic forgetting degrades a model's generalization capabilities by eroding previously learned information. More strikingly, we found that under certain conditions, continued adaptation is actively counterproductive, yielding worse results than if adaption had stopped. This is because as we continue to adapt, we forget more than we learn.
>
> The table below illustrates this clearly. The standard TDA model's performance peaked after 20,000 samples and then degraded. A model whose training was frozen at that point (TDA frozen) consistently outperformed the continuously trained model thereafter. Our proposed method, DOTA, effectively mitigates this degradation and delivers robustly superior performance.
>
>
>
> | Samples Seen            | 5k | 10k | 15k | 20k | 25k | 30k | 35k | 40k | 45k | 50k | Ave |
> |-|-|-|-|-|-|-|-|-|-|-|-|
> | TDA                     | 0.14     | 0.7       | 0.76      | 1.22      | 0.96      | 0.98      | 1.02      | 0.68      | 0.46      | 0.24      | 0.72        |
> | TDA(frozen after 20k) | 0.14     | 0.7       | 0.76      | 1.22      | 1         | 1.54      | 1.04      | 0.82      | 0.66      | 0.68      | 0.86        |
> | DOTA                    | -0.02    | 1.18      | 1.48      | 1.98      | 1.64      | 2.84      | 2.74      | 2.26      | 2.06      | 2.84      | 1.90        |
>
>
> [2] Catastrophic Interference in Connectionist Networks: The Sequential Learning Problem

---

> ### Author Response · Authors · 2025-08-06
> **Follow-up on our Response**
>
> Dear Reviewer,
>
> Thank you for your valuable feedback. In our author response, we have provided a detailed response to your concerns and questions.
>
> We hope our response addresses your concerns and would be happy to discuss any follow-up questions you may have.
>
> Thank you again for your time and effort!

---

> > ### Comment · Reviewer_J9Lm · 2025-08-08
> > **Follow-up Comments**
> >
> > I have no more questions. Thanks for the response. I maintain my score.

---

### Comment · Area_Chair_P5m3 · 2025-08-05
**Discussion with authors**

Dear Reviewers,

Please go through the author rebuttal and clarifications and let the authors know if you have any remaining concerns, so that they are able to respond timely. If the response is satisfactory, mention that too.

Also, please complete the Mandatory Acknowledgement only after the discussion.

Thanks,
AC

---

### Author Response · Authors · 2025-08-08
**Thanks!**

Dear AC and reviewers,

We would like to express our sincere gratitude to the reviewers for their valuable feedback and insightful discussions. We are also very grateful to the Area Chair for organizing the review and discussion phase. During the discussion, we have addressed all the questions raised by the reviewers, and the corresponding revisions will be incorporated into the final version of our manuscript. We find that all reviewers have confirmed that they have no further questions or concerns and have a clear understanding of our contributions.

Thank you again for your time and guidance throughout this process.

Sincerely

---

### Decision · Program_Chairs · 2025-09-17

**Decision:**

Accept (poster)

**Comment:**

This paper presents DOTA, where a vision-language foundation model (e.g., CLIP) is adapted to address distribution shifts during test time by continuously estimating the underlying test data distribution. The reviewers appreciated the motivation of the approach, its performance and extensive evaluation. All the concerns have been satisfactorily addressed in the author response. So the recommendation is to accept the paper.